# Contextual conditions define maximum energy-use threshold in low-carbon controlled environment agriculture for agri-food transformation

Shiwei Ng [1,2] ✉, Olaf Hinrichsen [1,2] & S. Viswanathan [3] ✉

Controlled Environment Agriculture has the potential to achieve food security and lower carbon emissions in agri-food systems. However, contextual factors such as what is produced and how it is produced determine the feasibility of meeting these goals. Here we show how the use of a Maximum Energy-use Threshold, shaped by these contextual factors, can define, identify and enable low-carbon operations. Results support the potential of low-carbon controlled environment agriculture over international import when growing leafy greens in land-locked countries with low grid emission factors or when substituting air freight of short shelf-life produce. Prospective low-carbon energy scenarios helps but optimising energy use remains critical. As controlled environment agriculture allows intensive farming with a reduced land footprint, controlled environment agriculture of high energy use crops as a lower-carbon alternative can be supported when the potential for agricultural land substitution and restoration for environmental services is considered, along with other contextual condition.

Controlled Environment Agriculture (CEA), which encompasses setups of varying degrees of automation, environmental control, and monitoring, holds many advantages over conventional open farming. Production scale and complexity of CEA can vary between different forms, from single storey vertical farms to multi-storeys plant factories with fully standardised and automated processes. It allows for intensive farming on non-agricultural land with reduced pesticide and water use, while being less vulnerable to climate change. Against the backdrop of more extreme weather patterns[1] and increasing competition for finite habitable land[2], CEA is primed to meet the global challenge of feeding an increasing world population.

Despite this, the extensive adoption of CEA faces headwinds due to high upfront capital and operational costs. As a result, the trickled-down cost to be absorbed by consumers limits commercial ventures to leafy greens and higher-value produce[3]. Unsurprisingly, because of the heavy use of artificial lighting and environmental control, rising energy costs have slowed operators' growth[4]. This intense energy use is the top carbon emission contributor of CEA[5,6]. It illustrates the current trade-off between energy, food security, and climate mitigation. With environmental merits being a key motivator for CEA adoption and perceived sustainability a main driver for consumer acceptance[7], lowering key performance indicators (KPIs), such as annual crop energy use-productivity (EUP), kWh/kg, is vital. With CEA-specific policies and standards continuing to take shape[8–10], KPIs that measure productivity and efficiency are important for benchmarking and disclosure purposes[11].

Climate change is one of the most pressing planetary boundaries that have been transgressed[12]. To achieve a lower carbon food system and possibly net zero emissions by 2050[13], environmental assessment of CEA for different siting scenarios is essential[14]. Often, environmental

[1]TUM School of Natural Sciences, Department of Chemistry, Technical University of Munich, Garching, Germany. [2]TUMCREATE, 1 CREATE Way, #10-02 CREATE Tower, Singapore, Singapore. [3]Nanyang Business School, Nanyang Technological University, Singapore, Singapore. ✉e-mail: shiwei.ng@tum-create.edu.sg; asviswa@ntu.edu.sg

impact analysis is obtained using life cycle assessment methodologies (LCAs)[15] and resource use is benchmarked with KPIs[16] (Fig. 1).

As different approaches to quantify sustainability, LCAs and KPIs are sensitive to contextual influences such as what produce is grown, how it is grown, where production is sited, and whether a low-carbon energy supply is available. Particularly, renewable energy availability has been shown to impact feasibility[17] and heavily influences eventual environmental impact[18]. Therefore, detailed LCAs with each siting scenario presenting their unique set of local conditions will conclude differently on whether benefits of common CEA impact proposition to local food systems could be realised. Often, impact propositions include sustainability claims such as CEA's potential to reduce or eliminate international transport emissions[19] and the potential to restore substituted land back into carbon sequestering native vegetation[20] as yield can be achieved on non-arable land, often in urbanised environment[21,22].

In this work, we develop and calculate a Maximum Energy-use Threshold, or MET, per kilogram of crop, for different countries and different future scenarios. This helps us in identifying, at an aggregate level, favourable scenarios that have the potential to realise the climate mitigation potential of CEA. For CEA ventures operating within this measure of operational prudence, a more detailed LCA evaluating the full environmental impact should then be embarked on as a separate extended sustainability assessment. The MET calculation considers the surrounding context through localised trade data[23], emission factors[24] and different shared socio-economic pathways[25,26]. The calculated threshold is fundamentally a balance between the anticipated CEA carbon footprint solely from energy use and the avoided carbon footprint implicit in CEA impact propositions (Fig. 2).

For CEA with estimates lower than this MET, they represent favourable siting scenarios that could accomplish a lower carbon food system if the impact propositions are achieved. LCA should be performed as a separate extension, which can then include emissions from other agriculture-related activities. CEA ventures with EUP higher than this threshold will negate any potential $CO_2$ emission reduction obtained by replacing an existing element within the local food system, even without performing an LCA.

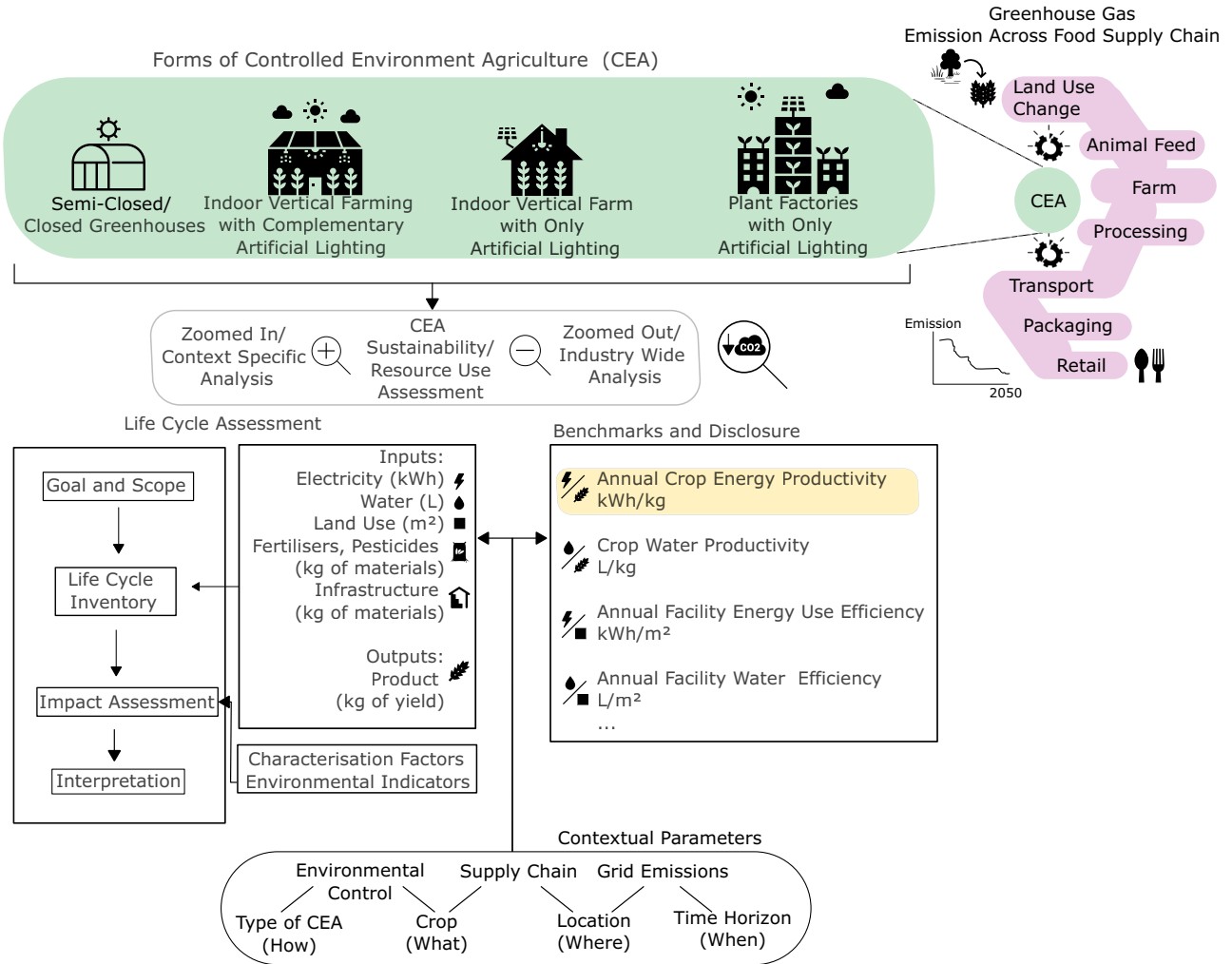

**Fig. 1 | Graphical overview of research gap.** This illustration highlights where Controlled Environment Agriculture, or CEA, has the potential to complement portions of our food supply chain and to better enable lower carbon food systems. Different forms of CEA are shown in the green bubble. For these different forms, two types of sustainability or resource use assessment commonly applied to CEA are shown. The magnifying glass symbol with a plus sign illustrates the zoomed in and context specific nature of life cycle assessment, while the magnifying glass symbol with a negative sign illustrates the broader perspective enabled by key performance indicators in benchmarks and disclosures. For both of these analyses, examples of contextual parameters impacting their results and relevance are shown. (Note: Wheat icon is licensed under an MIT license. Electric current and Drop symbol is designed by Freepik. Disruption symbol is designed by Afian Rochmah Afif · Freepik.com. ", " and " are all designed by Geotatah and licensed under an CC Attribution License. The logos were used without change, except the " where the sun icon has been removed.).

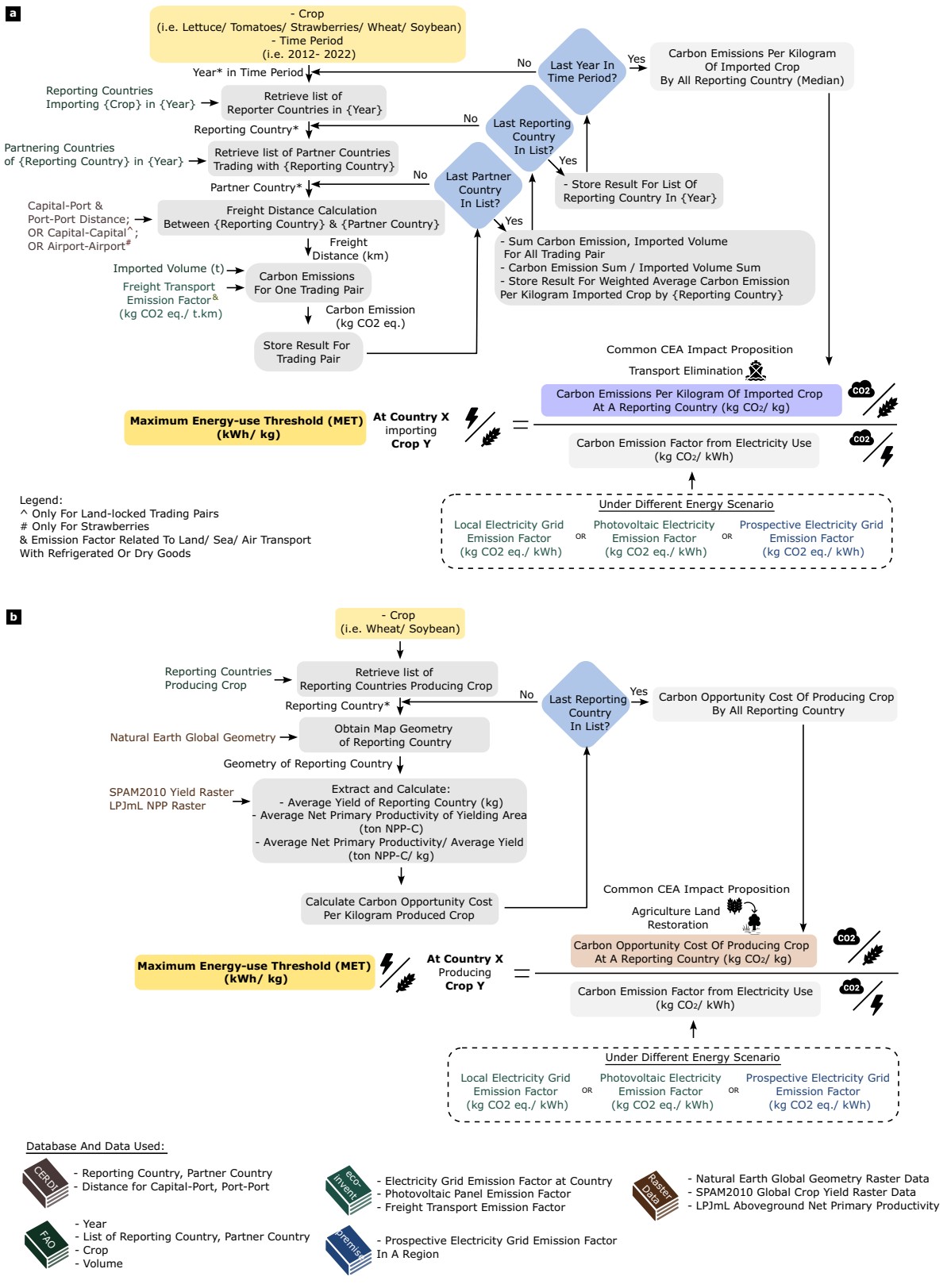

**Fig. 2 | Definition and flowchart for calculating Maximum Energy-use Threshold (MET). a** Flowchart of how MET is calculated when considering the impact proposition of transport elimination in the agri-food supply chain. The transport emission related to import is calculated by taking in data from multiple sources. Asterisks and values in cursive brackets represent variables that change at each iteration in a specified range. **b** Flowchart of how MET is calculated when considering the impact proposition of agricultural land restoration. Here, the carbon opportunity cost of existing crop-yielding land was calculated and used in the MET. The data used and their sources are highlighted using the book symbol with different colours. (Note: Wheat icon is licensed under an MIT license. Electric current symbol is designed by Freepik.).

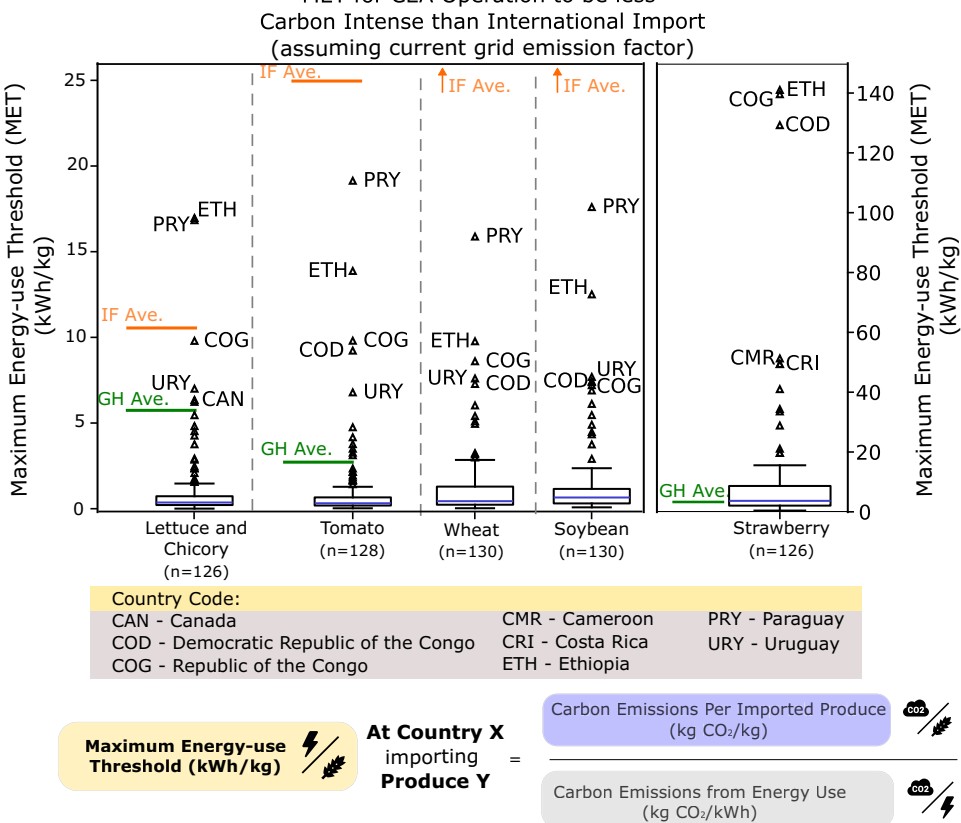

**Fig. 3 | MET considering combinations of contextual parameters.** The boxplot reflects the median (blue center bar), interquartile range (± 25% of median), whiskers (1.5 times the interquartile range), and outliers (triangle markers outside of whisker cap). The solid orange line represents the average energy use-productivity of indoor farms for respective crops. Orange, upward arrows denote that the average energy use-productivity of indoor farms is well above the highest MET and is not captured within the displayed range. The green solid line represents the average energy use-productivity of greenhouse for respective crops. All locations are identified with ISO3166 Alpha3 codes. Literature pertaining to indoor farming of strawberries and greenhouse cultivation of wheat and soybeans was not found. Hence, wheat and soybean do not have an average estimate for greenhouse, while strawberry does not have an average estimate for indoor farming. All countries, except the top five outliers, are not labelled for legibility. Full data are provided in Supplementary Data 1. The factors taken into consideration when calculating the MET are shown below the country code legend. (Note: Wheat icon is licensed under an MIT license. Electric current symbol is designed by Freepik.).

Because the MET is calculated based on external contextual factors such as emissions from the current supply chain (the incumbent option) and the local electricity grid, the relevance of the MET is not constrained to a specific form of CEA and does not vary depending on the equipment used. Current EUP estimates of CEA are compared with these contextualised MET. In doing so, the aim of this article is to determine the contextual conditions that can potentially allow CEA to mitigate Energy-Food-Climate Trade-off, before heavy investment of time and resources into an LCA. The analysis here explores (1) present circumstances considering international import substitution, (2) prospective low-carbon energy scenarios, and (3) present and prospective circumstances considering the opportunity for land use restoration.

## Results

First, for present circumstances considering international import substitution, growing leafy vegetables or short shelf-life fruits using CEA in land-locked, low-grid-emission nations are a few scenarios potentially presenting a favourable energy-food-climate trade-off.

Figure 3 shows the calculated MET for different countries based on their current grid emissions and trade patterns. The nations analysed here import selected produce of interest and report to FAO[23]. These crops – viable in both semi-closed greenhouse setups and indoor farms with artificial lightings – include commercially favourable leafy greens, vine vegetables, and fruits[27]. High calorific value crops,

such as wheat[28,29] and soybean[30], are analysed as well because they are extremely important as staple foods.

A large number of the countries have a low MET as a result of either a low-carbon emission from existing imports or high carbon emissions from existing electricity supply. This means that a CEA operation has to be extremely efficient in energy use, per kilogram of crop, to be less carbon intensive than current import. The average EUP for different crops and forms of CEA was extracted from literatures to visualise how current CEA compares to the METs. Countries with MET above these markers denote a favourable scenario for CEA implementation today, given the current operational state of CEA (GH – Greenhouse, or IF – Indoor Farming).

The estimates of indoor farming configurations exceed the MET allowable in growing the analysed produce for almost all countries. The only exceptions to this are a few scenarios for growing lettuce. This signifies that indoor farming, at its current development, will likely not enable a lower carbon food system as compared to current food imports. However, in the case of strawberries, where the produce is assumed to be air freighted due to short shelf life, possible aversion of higher emissions from air transport will allow higher MET of strawberries or other air-freighted produce in CEA.

In this analysis, countries such as Ethiopia, Congo, and the Democratic Republic of Congo are outliers and favourable siting locations for siting indoor farming because of their exceptionally low

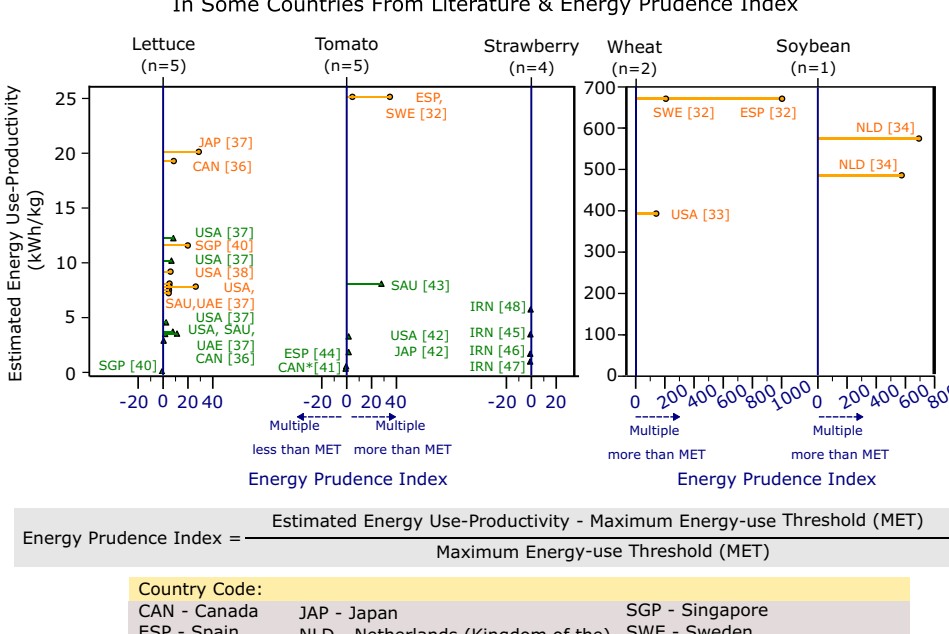

**Fig. 4 | EUP estimates from literature and the energy prudence index.** The vertical dot chart illustrates the estimated EUP of both greenhouses and indoor farming setups in various countries from the current literature. Greenhouse benchmarks are colour-coded in green, while indoor farming benchmarks are colour-coded in orange. The estimate asterisked for growing tomatoes in Canada denotes the use of other energy forms for heating requirements. If heating requirements are fully electrical-based, estimates will be much higher. The blue line represents where EUP meets the MET perfectly. A positive displacement to the right signifies higher energy use to the MET, which is not desirable, and a negative displacement to the left signifies lower energy use to the MET. The equation to calculate the Energy Prudence Index is showed above the country code legend.

grid emissions from the high utilisation of hydropower and reliance on traditional biomass (charcoal, crop waste, and other organic waste) as primary energy source[31,32]. For Paraguay, the cost of being land-locked[33] and reliance on distant trade partners position them as favourable for siting indoor farming as well. Resonating with the results here, another study where indoor vertical farming feasibility scores are synthesized from development indicators, Ethiopia scored favourably to host indoor vertical farming in macro indicators such as food security, climate change vulnerability, infrastructure, and others[34]. Though externalities such as grid stability and capacity can be a concern in Ethiopia, siting CEA can be possible as energy-intensive technology infrastructure like data centres already exists and provides a template for planning high energy use, high cooling load ventures[35].

Delving deeper into the literatures that informed the current average energy use of GH and IF here, the energy use estimates of individual literature are presented in Fig. 4. The literatures presented a wide range of EUP metric for lettuces[36–40], tomato[28,41–44], and strawberry[45–48].

Interestingly, growing lettuce in a semi-closed greenhouse at locations with very cold climates, such as Minnesota (USA) and Washington (USA), can perform worse than fully indoor farms due to inadequate solar irradiation, high heat loss, and consequently, high heating requirements[49]. To evaluate the climate mitigation potential of CEA in a particular country, an "Energy Prudence Index" is suggested here. A positive value in the Energy Prudence Index signifies a higher energy use than the MET, and a negative value in the index signifies a lower energy use than the MET.

Based on literature, almost all of the current CEA setups operate at EUP several multiples of the MET. Only in setups where extremely low energy is used and where short shelf life produce such as strawberries are air freighted, their operations hold potential to be a lower carbon option. For example, utilising a hydraulic rotating greenhouse setup in Singapore for CEA of lettuce can be achieved at an EUP of 0.021 kWh/kg[40], lower than the MET (Supplementary Data 1). As a land-scarce nation intensifying agri-food industry investments to bolster food security[50], it will be beneficial for Singapore to invest in low-energy greenhouses for lettuce cultivation.

Next, when considering prospective, low-carbon energy scenarios, the need for operational prudence as reflected in the MET relaxes but do not negate the need to further optimise CEA efficiency.

As solar power gains momentum to be the dominant form of renewable energy globally[51], and energy use emissions can be reduced through other forms of renewable energy, the MET was investigated under two main prospective energy scenarios here: (1) CEA facilities fully powered by current 3 kW$_{peak}$ photovoltaic panel technology or, (2) with the projected regional grid emissions factor for 2050 as electricity grids incorporate more renewable energy.

For land-locked countries, such as Paraguay, that possessed some of the most ideal energy grid and trade conditions for siting CEA today, the transition to solar energy presents a more stringent operating condition, as reflected in a much lower MET. This is due to photovoltaic technologies possessing a higher emission factor than hydropower (Supplementary Fig. 1). On the other hand, many countries with national grids of higher emission factors can benefit from higher photovoltaic utilisation, as reflected in the widening of the interquartile range and higher global median (Fig. 5a).

In addition to using photovoltaic panels, there are many pathways to achieving a lower emission factor energy supply. As nations pledge to transition to more renewables and towards a lower radiative forcing level in 2050[52], a myriad of possible grid emission factors are possible.

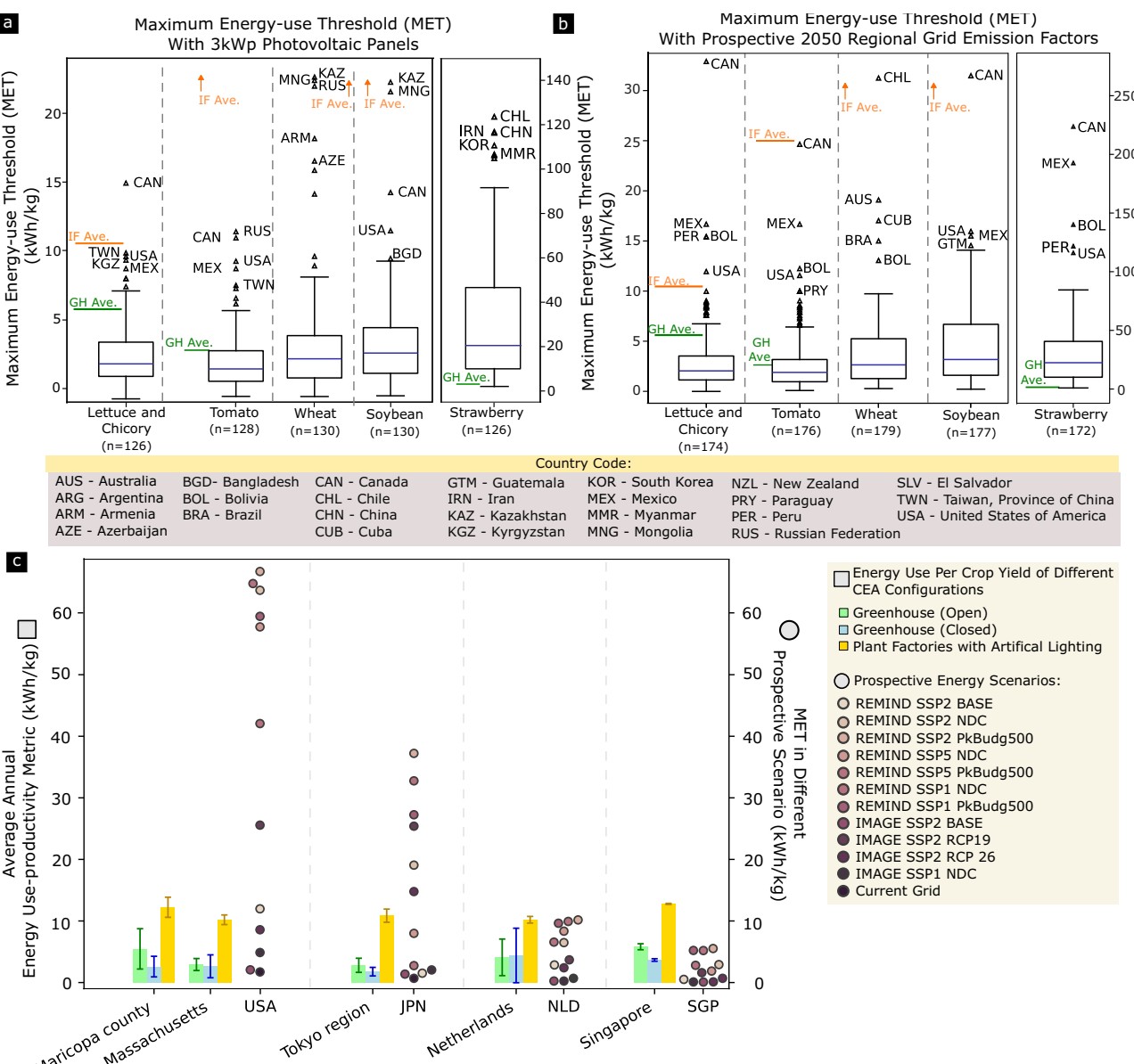

**Fig. 5 | METs under different energy scenarios and the feasibility of CEA in selected regions of interest under prospective energy grids. a** MET for produce-importing countries if CEAs are powered by monocrystalline silicon photovoltaic panels[24]. Here, the emission factor of low-voltage electricity production using flat roof installation was applied to all countries. The boxplot reflects the median (blue center bar), interquartile range (± 25% of median), whiskers (1.5 times the inter-quartile range), and outliers (triangle markers outside of whisker cap). Solid orange line represents the average energy use-productivity of indoor farms for respective crops. Orange, upward arrows denote that the average energy use-productivity of indoor farms is well above the highest MET and not captured within the displayed range. Green solid line represents the average energy use-productivity of green-house for the respective crops. Wheat and soybean do not have an average estimate for greenhouse, while strawberry does not have an average estimate for indoor farming. **b** MET for importing countries under prospective energy grids as nations shift towards increasing renewable energy composition. The boxplot reflects the median (blue center bar), interquartile range (± 25% of median), whiskers (1.5 times the interquartile range), and outliers (triangle markers outside of whisker cap). Prospective energy grid developments were assumed to follow a "middle-of-the-road" shared social-economical pathway, SSP2 Base, projected using the REMIND Integrated Assessment Model. **c** Extracted from literature, the error bar graph represents EUP estimates of growing lettuce using CEA under different climatic conditions[17]. The graph presents the mean and standard deviation of the EUP over twelve months (*n* = 12). The swarm plot presents the MET for the respective countries. Each MET under different prospective energy scenario is represented by a dot data point (circle symbol).

Here, the scenario "SSP2-Base" was assumed, where the shared socio-economic pathway (SSP) was extrapolated from historical patterns and possesses a base emission target causing a change in energy balance equivalent to 6.0 W/m² [25,53]. This positioned countries differently from earlier results as outliers favourable for CEA (Fig. 5b). It is important to note that while this approach[26] allows estimated emission factors to be better aligned with outputs from climate change integrated assess-ment models (IAMs), several nations in the adjusted life cycle inventory database can inherit higher grid emission factors as a result of geographical aggregation in regional outputs. As an example, for countries already possessing low grid emissions, such as Paraguay, higher emission factors from the Latin American region in REMIND IAM were inherited[54]. As a result, the respective MET was lowered, reflecting a more stringent operational requirement. Hence, for these countries, the present is already the best-case scenario. Energy pro-jection scenarios in 2050 can be refined with greater granularity by

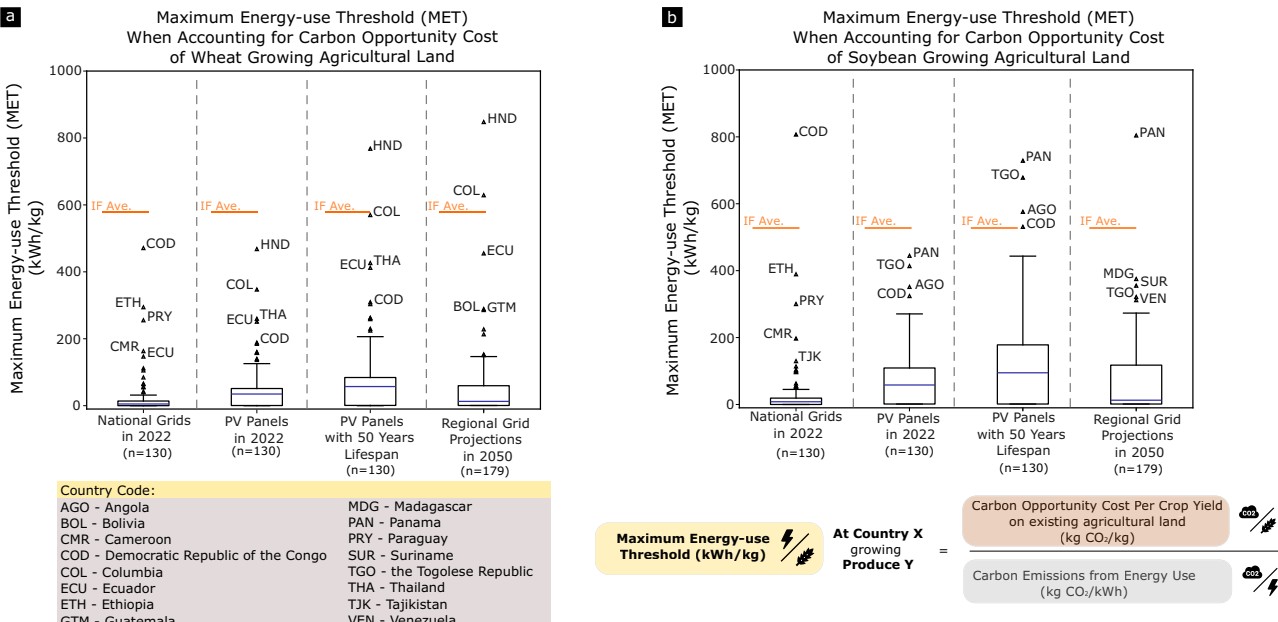

**Fig. 6 | MET of wheat and soybean producing countries when considering the carbon opportunity cost of agricultural land use. a** The boxplot reflects the median (blue center bar), interquartile range (± 25% of median), whiskers (1.5 times the interquartile range), and outliers (triangle markers outside of whisker cap). Solid orange line represents the average energy use-productivity of indoor farms for respective crops. Wheat and soybeans do not have an average estimate for production in greenhouses. Here, in addition to the three energy scenarios described in prior sections, a scenario with an extended technical lifespan of photovoltaic panels was added to gauge the impact of circular developments in solar energy. **b** MET for soybean-producing countries. The boxplot reflects the median (blue center bar), interquartile range (± 25% of median), whiskers (1.5 times the interquartile range), and outliers (triangle markers outside of whisker cap). Similar to wheat, CEA of soybean could be viable for selected countries. The factors taken into consideration when calculating the MET are shown below panel. (Note: Wheat icon is licensed under an MIT license. Electric current symbol is designed by Freepik.).

downscaling IAM results for a better estimate for the individual countries[55]. The METs considering alternative SSPs and IAMs are appended (Supplementary Fig. 2). To better illustrate the effect of different SSP, a subset of this comparison pertaining to CEA of lettuce is shown in Fig. 5c. Among the alternative SSP scenarios analysed, "SSP2" presented a more conservative MET as compared to other SSP considering different levels of barriers to mitigation and adaptation. Likewise, as more stringent emission targets are simulated, the prospective grid emission factor reduces and relaxes CEA operation requirements through a higher MET. IMAGE IAM is included here for sensitivity analysis[56].

Multiple scenarios of CEA configurations operating in different SSP could contribute to both food security and sustainability objectives. For example, plant factories operating in Massachusetts or Maricopa County could be viable if global climate initiatives limit global warming below 1.5 °C as the world "takes the middle road" (SSP2) or Nationally Determined Contribution (NDC) targets are achieved as the world "takes the green road" (SSP1). For other selected countries that have interest in CEA implementation, the METs across different prospective scenarios are appended (Supplementary Fig. 3). To compare EUP variance of different CEA configurations (plant factories and greenhouses) operating in different climate conditions for selected regions of interest, findings from a previous study were extracted[57] (Supplementary Fig. 4). Indoor farming configurations have higher EUP requirements but show less variance in different climate conditions.

Beyond leafy crops, cultivating higher calorific value crops such as wheat and soybean under current or prospective energy scenarios will not achieve a lower carbon food system as compared to importing. However, there is an additional dimension to conventional agriculture in which CEA provides plausible environmental benefits. Cereal crops require more land than other crops. Growing cereals such as wheat requires more than four times the arable land than tomatoes[58]. For these crops, the potential restoration of land previously used for agriculture[59] or preventing further land use change through CEA can avail originally arable land for climate mitigation purposes[60].

When considering such scenarios, CEA of cereal crops can possess climate mitigation potential in a few countries. Just as carbon emission occurs because of direct land use change, carbon sequestration can occur if agricultural land is reverted for native vegetation growth[61]. Here, a carbon opportunity cost is valued to account for this opportunity of carbon sequestration on land currently occupied to produce wheat and soybean[20]. The countries analysed are wheat and soybean producing nations as reported to FAO[23]. The result in Fig. 6 shows the MET of wheat and soybean CEA, if the CEA production is able to substitute agricultural land outputs and that land can now be restored to provide environmental services.

The current average energy use of wheat and soybean CEA is above the maximum threshold for almost all countries, signifying that CEA in its current development will not achieve a key sustainability objective of climate mitigation if such high calorific value crops are grown; However, there are a few outlier countries presenting unique scenarios where CEA could mitigate this trade-off. Tropical countries such as the Democratic Republic of Congo, Honduras, and Colombia possess some of the highest net primary productivity of natural vegetation and lowest grid emissions. They stand to be favourable CEA locations for wheat (Fig. 6a), while countries such as Panama could benefit from CEA of soybean (Fig. 6b).

Extending the technical lifespan of photovoltaic panels is critical, as it places lesser demand on virgin material and produces lesser waste[62]. Hence, improving the reliability and durability of photovoltaic panels towards a 50-year technical lifespan module is essential[63,64].

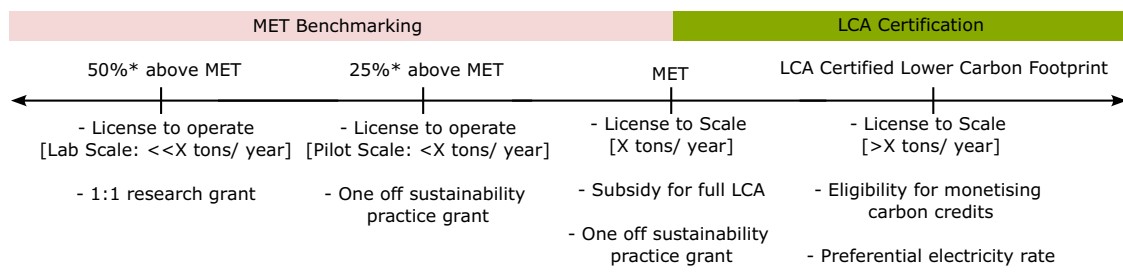

**Fig. 7 | Possible ways to institutionalise MET to encourage sustainable controlled environment agriculture.** The two-colour banners denote operational range where MET or LCA is more relevant. The asterisk symbol denotes a percentage that policymakers can adjust or add more increments in between. Some of the suggested policy levers are listed below each milestone.

Here, the emission of a photovoltaic panel kept in operation for 50 years was estimated to reflect this technological outlook on photovoltaic panel development. To calculate the emission factor of a 50-year technical life span solar panel, the existing life cycle inventory of a 30-year technical life span solar panel was extended to continue at a degrading annual yield for another 20 years. Because of the prolonged lifespan, the total yield increased, and its emission factor (kg $CO_2$ eq./kWh) was reduced. This further relaxed the MET as compared to calculations using present day emission factor.

While this article focused on contextual factors, the conclusions will also be influenced by methods for carbon accounting and carbon opportunity cost accounting. In the above calculation, MET is calculated on the accounting basis of "territorial" emission, where the carbon opportunity costs were only attributed to the producing nation. However, as nations do not consume the total production and can import in addition to what is produced, carbon opportunity costs can also be adjusted for trade. To this end, MET calculated on the basis of consumption across different prospective energy scenarios is presented in Supplementary Fig. 5. In the carbon opportunity cost accounting method, in which spared agricultural land is assumed to be restored, a historical accounting perspective can also be taken, where carbon loss from agricultural land expansion occurring between Year 2000 and 2010 was used (Supplementary Fig. 6). From both accounts, outliers can present much more relaxed MET requirements.

## Discussion

Here, the Maximum Energy-use Threshold (MET) per kilogram of crop is introduced as a resource use metric balancing energy-food-climate considerations for CEA. It is a tool complementary to environmental impact assessment methodology such as LCA, where LCA could include impacts from other sources, such as infrastructure. LCA remains important, as some forms and scales of CEA might require little energy, but with most of its emissions stemming from infrastructure – such as a small-scale urban farm[65,66]. Here, because energy use is a key carbon emissions contributor in larger-scale CEA, the MET provides a measure of prudence where operation exceeding it will not position CEA to achieve a lower carbon food system, even without conducting an LCA. Operating below this measure of prudence places a CEA scenario in good standing for a more detailed analysis using LCA to confirm its full environmental impact profile.

An objective and transparent benchmark is necessary to guide the development of CEA. Objective in being independent of a CEA operator's potential bias during its calculation and being transparent in its data and approach. The MET can add value by informing operators of an objective benchmark to optimise towards and for policymakers to objectively form policies that incentivise sustainable operation. Elaborating further, here are the two ways in which the MET can already be valuable.

First, for identification of prospective CEA sites. As detailed in this work, the CEA of leafy crops in land-locked nations with low grid emissions can already be advantageous compared to international imports, and for more countries when considering prospective energy grids. Similar to the potential of relocating agricultural activities[67], it can be viable for collaborating nations with regional trade agreements to perform CEA in selected countries to bolster regional and national food security.

Next, the MET could serve as an industry benchmark for determining financial incentives by policymakers in CEA-specific regulations. The MET is technology agnostic in its formulation and could be an attractive proposition to policymakers because they form a target for the CEA industry to rally towards. On a practical level, different forms of CEA will find meeting the target a different challenge. Operators will have to choose wisely the appropriate level of automation and environmental control needed to obtain an optimised yield from a target crop. Imprudent implementation of plant factories in every scenario will make meeting MET extremely difficult. From the perspective of a CEA venture or researcher, the criterion provides an aspirational target to optimise their operation to minimise carbon footprint.

Beyond this first calculation of MET for different countries, more granular data will be needed to quantify MET more precisely and avail more specific policy use cases. Validation can be feasible with national, state, or province-level trade data obtained through local agencies, instead of relying on aggregated data, thereby producing a more precise quantification of the MET at a regional level. While MET cannot be used to certify the sustainability of a CEA operation, policymakers can use MET to promote sustainable practice of CEA.

Figure 7 illustrates one possible way to do this. For operations above the MET, where large-scale CEA is highly unlikely to be sustainable even without performing an LCA, such unsustainable practices should be limited in production volume; Allowing CEA operators to scale up in production volume only upon demonstrating an improvement. Incentives can come in the form of co-funded research grants, one-off sustainability practice grants, or subsidies to perform full LCA. Only when operators are below the MET and certified through an LCA that it is a lower carbon option, preferential electricity rate or access to carbon trading as a seller can then be made available. Consensus building with operators will be critical in determining parameters such as the appropriate MET milestones and production volume limitations to not deter new entrants or stifle small start-ups and instead motivate sustainable innovation practices. Here, the approach and formulation to calculate a MET based on contextual conditions paves the way forward for such strategies.

It is worth highlighting that a particular CEA system operating below the MET does not confirm the accomplishment of CEA impact propositions such as international import elimination or agricultural land restoration. These agrifood system transformations will require not just technical innovation in CEA operation but also supporting social-economical and policy conditions involving diverse stakeholders[68]. Particularly for the link between agricultural

**Table 1 | Emission Factor for Transportation calculation and applied scenario**

| Ecoinvent 3.9.1 Unit Process | Emission Factor (kg. $CO_2$ eq. per ton.km) | Applied to: |
|---|---|---|
| market for transport, freight, sea, bulk carrier for dry goods | 7.08E-03 | Crop: Wheat, Soy<br>Mode: Maritime Transport |
| market for transport, freight, sea, container ship with reefer, cooling | 2.27E-02 | Crop: Lettuce, Tomato<br>Mode: Maritime Transport |
| market for transport, freight, aircraft with reefer, cooling | 0.837405383 | Crop: Strawberry<br>Mode: Air Transport |
| market for transport, freight, lorry, unspecified | 1.53E-01 | Crop: Wheat, Soy<br>Mode: Land Transport |
| market for transport, freight, lorry with reefer, cooling | 1.43E-01 | Crop: Lettuce, Tomato, Strawberry<br>Mode: Land Transport |

intensification through CEA and land sparing, stimulative conditions are required to promote land sparing for environmental services[69–71]. They include policies for conservation set-aside programs with financial instruments to remunerate landowners for the alternative use of rural lands[72–74]. The holistic evaluation of sustainability encompassing ecological, social, and economic aspects will require other conceptual frameworks[75–77], and presents opportunities for future work.

The sustainability assessment of CEA, as well as other emerging technologies, is highly nuanced and can often cause unintended rebound consequences. For example, as CEA reduces agricultural land use, there can be more arable land available, given the assumption that stimulative conditions for land sparing are present. These stimulative conditions will include financial incentives that can be more attractive than the gains from using the land for food production. This can cause an increase in the cost of goods for many sectors, threatening the promise of enhancing food security.

Alternatively, if CEA were to be hosted on non-arable land, social rebound effects can emerge. For example, to reduce CEA thermal cooling load and the cost of operation, CEA operators favour locations with cool, temperate climates and low electricity costs. Often, these places would already have communities settled there. A parallel to this rebound effect can be drawn from the dense siting of data centres in North Virginia, USA[78]. Quality of life impact, rising local utilities bills, water scarcity, decreasing property value, and declining local community support have been reported.

The environmental consequence of substituting traditionally farmed products with CEA products demands a separate and far more complex analysis. Consequential life cycle assessment, which considers the implications on marginal suppliers due to a change in supply and demand, can be a first step[79]. Modelling stakeholders as agents and integrating emergent behaviours from their interaction into life cycle assessment models can be the next step[80–82]. The environmental impact and benefits of CEA, as well as emerging innovations, are highly nuanced. Advanced modelling tools such as LCA will no doubt be necessary to aid decision making[83,84]. Here, the MET provides a complementary benchmark in CEA before such detailed modelling and analysis is embarked on.

## Methods
### Estimation of Transport Emissions
Agri-food supply chains are heterogenous and can change due to a myriad of factors[85], and the associated transport emissions varies due to transport options, year of assessment and surrounding geographical condition. To calculate the Maximum Energy-use Threshold, emissions arising from importing different crops was first calculated.

First, in a given year, every pair of reporting import nation and the partnering export nation were identified with their respective trading amount. This was extracted from the detailed trade matrix data published by Food and Agricultural Organisation (FAO), between a time period of 2012 to 2022[23]. Next, the produce were assumed to be transported between the capital of these trading pairs. Bilateral sea

distance between the capitals of reporting and partner nations was obtained from publicly available CERDI sea distance database[86], while the air freight distances were calculated with geodesic between obtained airport coordinates from public domain[87]. This included road distances from their capital to respective nearest ports, airports. For trading pairs where both countries are land-locked – without access to ports and directly accessible to each other overland – the shortest road distance between capitals was used instead[86]. The resulting measure of freight transport, ton.km, for associated land and maritime transport was calculated by multiplying the traded amount and respective transportation distances (Eq. 1).

$$Freight\ Transport_{a \to b, crop}(tons.km)$$
$$= \begin{cases} M_{import} \times \left( d_{a,cap \to a,port} + d_{a,port \to b,port} + d_{b,port \to b,cap} \right) \\ M_{import} \times \left( d_{a,cap \to b,cap} \right), & \text{if both countries are land} - \text{locked} \end{cases}$$
$$(1)$$

Subsequently, to calculate the carbon emissions, $CE_{a \to b}$, attributed to that trading pair (kg. $CO_2$ eq.), the measure of freight transport, ton.km, for both land, maritime and air freight was multiplied with the emission factors (kg. $CO_2$ eq. per ton.km) associated with the respective mode of transport[24].

$$CE_{a \to b, crop}(kgCO_2 eq.)$$
$$= \begin{cases} M_{import} \times \left[ \left( d_{a,cap \to a,port} + d_{b,port \to b,cap} \right) \times ef_{land,crop} + \left( d_{a,port \to b,port} \times ef_{sea,crop} \right) \right] \\ M_{import} \times \left( d_{a,cap \to b,cap} \right) \times ef_{land,crop} & \text{, if both countries are land} - \text{locked} \end{cases}$$
$$(2)$$

For lettuce, tomato, and strawberry, refrigeration were assumed to be required during freight transportation while wheat and soybean were assumed to be dry goods (Table 1).

Because a single importing nation can have multiple trading partners supplying a produce in a given year, carbon emissions from all associated trading pairs are summed (kg. $CO_2$ eq.). The total imported amount of produce from all associated trading pairs are summed as well (kg) and used to reciprocate total carbon emissions.

$$CEI_{import,c,year} = \frac{\sum_{n=Partners,year} CE_{a \to n, crop}(kgCO_2 eq.)}{\sum_{n=Partners,year} M_{import_{a \to n, crop}}(kg)} \qquad (3)$$

This resulted in the weighted average of carbon emissions per kilogram of imported produce (kg. $CO_2$ eq. per kg produce) for a particular year. This is iterated over all reporting, import nation for all the analysed crops. Because import quantity and partner changes due to a myriad of reasons, the calculation of this weighted average iterated each year, for a time period between 2012 and 2022. To have a representative value over these ten years, the median was taken. To derive the Maximum Energy-use Threshold (kWh per kg), the median carbon emissions per kilogram of imported produce was divided by

the current and prospective electricity emission factor (kg. $CO_2$ eq. per kWh).

Due to the unavailability of actual shipping and air freight data, validation of sea freight distance and air freight distance was not feasible here. However, the shortest straight-line approach using a geo-disc here is consistent with the shortest straight-line approach using a Great Circle Distance by the International Civil Aviation Organisation in its freight carbon emission calculator[88].

Here, it was assumed that even as total produce demand and supply chains can change over time in prospective scenarios, such changes do not cause material change from the calculated value.

### Prospective Energy Scenario: Increasing technical lifespan of photovoltaic panels

To estimate the emission factor of a 50-year technical lifespan photovoltaic panel, the life cycle inventory of *"Electricity, low voltage {RoW}| electricity production, photovoltaic, 3 kW$_{peak}$ flat-roof installation, single-Si | Cut-off, U"* was adjusted. In the original inventory, the carbon emission of producing 1 kWh was due to the production of photovoltaic panel. To attribute the emissions to 1 kWh of generated power, the total emissions was reciprocated by the total produced power of photovoltaic panel in its lifetime. There, the panel was assumed to have a 30-year lifetime of 3 kW$_{peak}$ capacity and produce power at an average annual yield of 1099 kWh/kW$_{peak}$[24].

Here, to simulate a 50-year technical life span, it was assumed that following a 30-year life span with an average annual yield of 1099 kWh/kW$_{peak}$, continuing use of the same solar panel is projected with a yield degradation rate of 0.5%/ year for mono-crystalline silicon for another 20 years[89]. The emissions attributed per kWh is calculated as the reciprocal of the total yield over its 50 years technical lifespan. The total yield over its 50 years technical lifespan was calculated as:

$$\left(30\,years * 3\,kW_{peak} * 1099\frac{kWh}{kWp} * 0.92\right) + \sum_{n=1\to20}[1099 * (0.995)^n * 3 * 0.92]$$
$$= 148575.72\,kWh$$

(4)

Referencing a study where the actual yield was lower than the report IEA data[90], a correction factor of 0.92 was adopted by Ecoinvent[24] and used here as well. The resulting emission factor, 0.0517 kg $CO_2$ eq. per kWh, was calculated using the IPCC 2021 GWP100 method on LCA software, SimaPro.

The percentage reduction in emission factor because of extended life cycle is similar to another LCA where End-of-Life scenarios of Photovoltaic Panels in Australia are conducted[91]. Here, the emission factor is higher than the reported 0.035 kg $CO_2$ eq. per kWh for a 50-year life span panel in Australia due to the life cycle inventory of the "Rest of World" region. In the "Rest of World" region, the inventories include contributions from more carbon-emitting countries.

### Prospective Energy Scenario: Projected Grid Emissions of REMIND SSP2 Base Scenario

To estimate the projected grid emission factor of different countries, the methodology of *premise* was replicated to modify the Ecoinvent 3.9.1 database[26]. Projections in 2050 were taken to follow a "Middle of the road" shared socioeconomic pathway, SSP2-Base, where trends do not change extensively from historical patterns[25], and emissions continue to grow at the same rate as the past. With these narratives, Integrated Assessment Models (IAM) IMAGE and REMIND projects what is the composition and efficiency of electrical grids in 2050. The *premise* open-source code was implemented, in which efficiencies of power generating technologies were updated, new life cycle inventories were created, and regional electricity grids were aligned to IAM geographical definitions. The resulting, regional emission factors were

used for all countries aggregated under it. Here, while IMAGE offers a relatively finer geographical aggregation, REMIND outputs was used due to a larger emphasis on photovoltaic energy, with lesser emphasis on hydropower and nuclear power.

### Quantifying the carbon opportunity cost of continual agricultural land use

To estimate the carbon opportunity cost incurred by continual agricultural land use, the "carbon gain" method proposed by Searchinger et al. was adopted for wheat yielding and soy yielding land[20].

First, to identify global crop-yielding land, *SPAM2010* cropland raster data was used[92]. The open-source software, *rasterio* was used to open and reproject all other raster data downloads to cropland raster data[93]. To obtain the wheat or soy yielding land of only selected countries, cultural vector data were downloaded from public domain, *Natural Earth*[94], and geometrical information about national boundaries were extracted using open software, *geopandas*[95], to clip cropland raster data. This was applied to net primary productivity of native vegetation (NPP$_{nat}$ per hectare) raster data and carbon stocks of potential natural vegetation (ton C per hectare) raster data[20], as well as terrestrial biome raster data[96].

In the "carbon gain" approach, to calculate the carbon sequestration by native vegetation of potentially, restored agricultural land; First, the mean NPP$_{nat}$ considering every cell/area in each producing country was divided by their respective average yield, resulting in a NPP$_{nat}$ per crop yield for each country. Referencing the calculation done by Searchinger et al., 0.5 tonne C per hectare per year of carbon sequestration was assumed to be generated for every one tonne of NPP$_{nat}$ available. For comparison, as suggested by Searchinger et al., 0.42 tonne C per hectare per year of carbon sequestration could be assumed for the tropics by dividing the average NPP$_{nat}$ of tropical croplands by the average carbon flux in tropics over 100 years. To convert kg C/kg crop yield to kg $CO_2$/kg crop yield, the result was multiplied by a factor of 3.67, which represents the molecular weight ratio of carbon dioxide to carbon.

$$Carbon\ Sequestration_{carbon\ gain}\left(\frac{kgCO_2eq.}{kg}\right)$$
$$= \frac{\sum NPP_{nat}(\frac{tonC}{ha})}{\sum Yield_{Area}(\frac{ton}{ha})} \times 0.5\left(\frac{tonC}{ha}\right) \times \frac{44}{12} \times 10^{-3}$$

(5)

### Reporting summary

Further information on research design is available in the Nature Portfolio Reporting Summary linked to this article.

## Data availability

The raw data and generated data in this study have been deposited in the Code Ocean database under the following link: https://doi.org/10.24433/CO.2717777.v1. The estimated energy use data for plant factories and greenhouses under different climatic conditions are available upon request because it is the work published by Weidner, T. et al.[57]. Access can also be obtained at the original, cited publication from the original authors. Supplementary Fig. 1 to Supplementary Fig. 6, and Supplementary Data 1 are available as Supplementary Materials.

## Code availability

Scripts performing the computation are written in Python programming language and are available at Code Ocean, under the following link: https://doi.org/10.24433/CO.2717777.v1. This includes all the calculation needed to calculate the Maximum Energy use Threshold and generate the results visualised in Fig. 3, Fig. 5a, b, 6a, b and swarm plot in 6c. Upon requests, the scripts creating the Supplementary Figs. visualisations will be made available as well. The

referenced data are the same output data available in Code Ocean after running the published code. The following software and packages are used: jupyter1.1.1; pandas2.2.3; numpy1.26.; country-converter1.3; countryinfo0.1.2; faostat1.1.2; gdal3.0.4; geopandas1.0.1; geopy2.4.1; kgcpy1.1.8; matplotlib3.9.4; pycountry24.6.1; rasterio1.4.3; rioxarray0.15.0; scipy1.13.1; setuptools57.5.0; shapely2.0.7; xarray-spatial0.4.0.

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

## Acknowledgements

This research is supported by the National Research Foundation, Prime Minister's Office, Singapore under its Campus for Research Excellence and Technological Enterprise (CREATE) programme through the grant entitled "Proteins4Singapore" [S.V. and O.H.].

## Author contributions

S.N. developed the methodology, conducted the analysis, and wrote the original draft. S.V. and O.H. aided in the conceptualization and development of the methodology, reviewed and edited the draft.

## Funding

## Competing interests

Authors declare no competing interests.
