## [Transparent Peer Review file · Nature Communications]

Contextual Conditions Define Maximum Energy-use Threshold in Low-Carbon Controlled Environment Agriculture for Agri-Food Transformation.

Corresponding Author: Dr Shiwei Ng

Version 0:

Reviewer comments:

Reviewer #1

(Remarks to the Author)

Controlled Environment Agriculture (CEA) production systems represent the most intensive form of food production. Thanks to their controlled and protected conditions, they offer a promising solution for food production, particularly in the face of climate change, water scarcity, and regions with degraded soils (soilless systems). However, these systems require significant energy and resource inputs, which could pose a sustainability challenge.

I found the concept, application, and analysis of the energy use-productivity threshold (MET) as a key performance indicator for CEA systems particularly interesting. This approach provides a valuable framework for assessing the sustainability of CEA systems in terms of energy and resource efficiency. The manuscript is well-written and well-structured. Attached to the revised/annotated PDF are some comments that may help enhance its quality and clarify certain points

(Remarks on code availability)

Reviewer #2

(Remarks to the Author)

First, thanks to the authors and the editor for the opportunity to review this manuscript. It was interesting and presented a novel way of thinking about spatio-temporal optimization for CEA. As I understand it, the manuscript aims to explore the tradeoffs between CEA energy use (and associated carbon footprint) and the limitations of traditional ag, including long-distance travel and expanding land use. Overall, I think this is a useful study for the Nature Comms audience, since it encourages folks to think spatially about this particular environmental challenge, and it offers general guidance on the types of situations that might justify extensive investment in CEA.

That said, I do have some comments that I think are important to address before the manuscript is ready to see publication. Overall, I think the article overstates the conclusions that can be drawn from these methods. This is not a test of low-carbon CEA – that would demand an LCA or other methods that includes inputs and infrastructure. I also think the human dimensions of the proposed interventions are largely overlooked and should be considered with more care. Finally, I think the figures and phrasing of some of the methods are difficult to interpret, and the authors should endeavor to make clearer connections between the actual numbers they calculated and the conclusions they draw. I would be happy to re-review it once these changes have been made. I list specific versions of these comments in no particular order for consideration:

1. “The calculated threshold is fundamentally a balance between anticipated CEA carbon footprint solely from energy use and avoided carbon footprint implicit in CEA impact propositions (Fig. 1b).” - I am concerned about the framing of CEA as environmentally friendly prior to any LCA work. As the authors themselves point out, there are a number of infrastructural investments that must be amortized as carbon per kilogram product – and other recent work in this family of journals has shown that infrastructure is perhaps the most important factor in LCA of urban ag (Hawes et al., 2024). There were similar findings in other, older CEA work (Goldstein et al., 2017). It might be more accurate to say that the operation of CEA could mitigate tradeoffs under confluent conditions, but I don’t think that the authors demonstrate clearly that CEA is low-carbon even if its operation is. This is ultimately a phrasing and framing issue, I certainly don’t expect the authors to conduct a full-

blown LCA to address this.

Hawes, Jason K., Benjamin P. Goldstein, Joshua P. Newell, Erica Dorr, Silvio Caputo, Runrid Fox-Kämper, Baptiste Grard, et al. "Comparing the Carbon Footprints of Urban and Conventional Agriculture." *Nature Cities*, January 22, 2024, 1–10. <https://doi.org/10.1038/s44284-023-00023-3>.

Goldstein, Benjamin, Michael Hauschild, John Fernández, and Morten Birkved. "Testing the Environmental Performance of Urban Agriculture as a Food Supply in Northern Climates." *Journal of Cleaner Production* 135 (November 2016): 984–94. <https://doi.org/10.1016/j.jclepro.2016.07.004>.

2. The manuscript should be edited once more to remove odd words or grammatical tricks here and there, like an extra "be" at line 77. "Transiting" at line 156. Sentence at 247. Etc.
3. In general, the figures could use some work. Aesthetically, they are quite nice, and they communicate a simple message quite well. But they fail to communicate the key message of the results section, which is where and under what circumstances the EUP exceeds or is below the MET. I don't know that I think the current figures need removed or changed, but an additional panel that shows the EUP divided by MET relative to 1 might be helpful.
4. At line 135, there is discussion of the low carbon impacts of electricity in certain developing countries. I have to wonder about the externalities associated with siting high energy-use crop growing systems in places where the grid may be unstable or unprepared for a new, large user. While I can appreciate the idea, it seems a bit short-sighted to me.
5. Related to this, I do not understand this statement – why is Ethiopia better than the others named? -- "Resonating with a previous study that assessed macro indicators for indoor vertical farming's feasibility and sustainability in Africa, Ethiopia stands out as the African nation suitable for CEA."
6. I am very unclear on what type of food is being offset in the agricultural land substitution scenario. Is this local food production replaced by CEA? Or imported food? If imported food, the explanation doesn't seem to make sense to me. The conclusion seems to indicate that this section is focused on how exporting countries might reduce their footprint – this is interesting and perhaps more internally consistent, but I have some concerns as discussed in my next point.
7. Overall, I remain a bit unconvinced of the idea that CEA inherently reduces the land use for agriculture in other locations. These sorts of telecoupling effects are unpredictable at best, and things like local labor markets are unlikely to simply vanish if the demand from a particular export partner is reduced. For that matter, the authors still need to talk about the in-migration to cities that would be required, the upskilling of agricultural labor, etc. If CEA requires less employees per kg of produce, it seems unlikely that the land use transition would be dramatic, since the original farmers still need a source of income.
8. The authors talk about an expanded technical lifespan for solar, but I don't think they ever say what 50 years is an alternative to. What is the technical lifespan in EcoInvent? This should be in the main text (I see it in the methods, but this is a Nature Comms piece, not all readers will keep going to find the methods).
9. I'm a bit confused in the methods section with Eqn. 1. If the country is land-locked, all food is assumed to be ground-transported? This doesn't make sense for, for example, import of bananas from Colombia to Switzerland. Some of that distance is, of course, by sea.
10. I appreciate that the authors try to differentiate between cooled lorry use and dry goods, but does Table 1 in the methods suggest that the carbon footprint of a refrigerated lorry is lower than that of an unspecified lorry? This might indicate that the "unspecified" option includes some combination of a variety of lorries, which wouldn't really be achieving the authors' goals.
11. The authors should be more specific about the programming languages used.

(Remarks on code availability)

I have not had time to replicate the code, but assume that the authors were careful and thorough.

Version 1:

Reviewer comments:

Reviewer #1

(Remarks to the Author)

General comments:

1. The MET concept is novel, but its relationship to existing environmental performance metrics (e.g., specific LCA indicators, energy use productivity benchmarks) could be more explicitly articulated. How does MET add actionable value beyond existing LCA thresholds?
2. The assumption that CEA systems meeting MET will inherently achieve climate benefits seems optimistic. A discussion on potential rebound effects or partial achievement of impact propositions would be helpful.
3. Figures 2–5 are rich in data but can be challenging to interpret. Consider simplifying axis labels, highlighting key outlier countries more consistently, and integrating policy-relevant thresholds directly into visuals
4. For trade distance assumptions: was any validation done to assess the realism of capital-to-capital distances vs. actual supply chains?
5. How were seasonality and perishability handled in emissions calculations (especially for leafy greens and strawberries)?

Specific comments:

- L3. Confluent: Maybe the term "contextual conditions" fits better
- L72: MET is an excellent construct, but its definition and derivation process could be clearer. Consider including a visual flowchart summarising its components (inputs, calculations, thresholds).
- L86-89: Have you deliberately excluded non-energy impacts in the MET (e.g., materials, embodied carbon of infrastructure)?
- L106 (Fig.3): The interpretation of MET vs. actual EUP is clear in figures, but some textual explanations could be more

concise and better linked to key implication

- L126-128: Could the authors clarify how air freight assumptions for strawberries were verified across the sample countries?
- L132: In the discussion of favourable locations (e.g., Ethiopia, Paraguay), have you considered local grid reliability and cold-chain logistics as potential limiting factors?
- L210 (Fig. 4): How sensitive are your MET results to PV efficiency improvements vs. emission factor assumptions?
- L293: In the discussion of favourable locations (e.g., Ethiopia, Paraguay), have you considered local grid reliability and cold-chain logistics as potential limiting factors? Other interesting issues that can be explored in the discussion section are:
 - a) Can the authors reflect more on how MET could be institutionalised — e.g., integrated into CEA certification, subsidies, or siting decisions?
 - b) What would be required to transition MET from research to policy use — what validation or consensus building is needed?

(Remarks on code availability)

Reviewer #2

(Remarks to the Author)

Thanks again to the authors, this time for conducting a thorough and effective revision. I remain convinced that this is a useful and interesting article, and I have very few comments to share. These are minor corrections that should not require further peer review. I look forward to citing this article soon.

1. I think the authors need to make a bit clearer in the main text what is considered in the MET. Specifically, I'd add a few sentences in the introduction or early results explaining how the carbon emissions per imported produce were found.
2. At line 77, "The calculation" should be clarified. Which calculation? MET?
3. A number of places need another grammar check. For example, "resource use are" at line 58 and "the" is missing in "within local food system" at line 85. There are other examples, just needs a careful copy-edit.
4. I think Fig 4 is missing some of the orange annotations that indicate where IF and GH fall on the MET spectrum.

(Remarks on code availability)

Version 2:

Reviewer comments:

Reviewer #1

(Remarks to the Author)

(Remarks on code availability)

General Response To Both Reviewers

We would like to thank both reviewers for their valuable comments and for providing us an opportunity to enhance the clarity of this work.

First, we are pleased that the work resonated with Reviewer 1 as a valuable framework for assessing CEA sustainability. Indeed, we hope this work informs the audience of a complementary way to think about evaluating CEA sustainability. The amended manuscript benefited greatly from Reviewer 1's comments to enhance on its clarity, particularly in the first results and discussion section.

Next, we thank Reviewer 2 for their valuable feedback and for giving many comments that has significantly improved the clarity and positioning of the paper. The reviewer has rightly distilled the spatial-temporal perspective that CEA innovators and policymakers should approach with, and that which we are communicating here. Without such a nuanced perspective to CEA, one might be quick to dismiss or overstate the potential in which it brings to transform the agrifood supply chain.

For ease of perusal, specific responses to reviewer 1 comments can be found from HERE, and specific responses to reviewer 2 comments can be found from HERE.

We thank both reviewers again for their insightful comments and welcome any further clarifications.

Point-by-point response to the reviewers' comments

Response to comments by Reviewer 1

➤ **Comment:**

Page 1, Line 14 – Amend “*the feasibility to meet these goals*” to “*the feasibility of meeting these goals*”

Response:

Thank you, done.

➤ **Comment:**

Page 1, Line 43 – Amend “*efficiency*” to “*efficient*”

Response:

In the original manuscript, the author used the wording to refer to the type of KPIs instead of the nature of the KPIs. To avoid confusion, the revised manuscript now reads:

“With CEA-specific policies and standards continuing to take shape, *KPIs that measure productivity and efficiency are important* for benchmarking and disclosure purposes”

➤ **Comments:**

Page 2, Figure 1, Panel A: “*how different are vertical farms with artificial lights and plant factories with artificial light? Please specify the different components that make these CEA systems differ*”

Response:

Though indoor vertical farms and plant factories are both closed environments with artificial lighting and environmental control, a subtle difference lies in the scale and system complexity. One of the ways to depict the difference in scale and complexity, as illustrated in Figure 1, is that plant factories often refer to facilities with multiple storeys of growing area, and with high level of automation and standardised process such as the use of conveyors and robotics. Currently, there is yet a formal classification that differentiates vertical farm from plant factories in terms of components in these two systems; also, the use of certain technology are rarely exclusive between the two forms of CEA. We do agree that this subtle difference is hard to pick up from the illustration. We added this clarification as the second line of the introduction right as we introduce CEA. It reads as:

“Production scale and complexity can vary between different forms, from single storey vertical farms to multi-storeys plant factories with fully standardised and automated process.”

➤ **Comment:**

Page 2, Figure 1, Panel B: *“in Figure 1b, I believe the first part of the graph is unnecessary since you present the impacting conditions and their relationship in the second part of the graph. The first part is a little confusing”*

Response:

We thank the reviewer for pointing out the potential difficulty in following the train of thought in the illustration.

The first part of the illustration shows the underlying common environmental impact proposition of CEA and how that benefit can be quantified in terms of global warming potential. The second part then rearranged the necessary balance between a CEA facility operation and the quantifiable benefits, in order for those propositions to be true. Hence, it shows the train of thought deriving the MET calculation. We considered the reviewer’s concern and opted to keep the first part of the figure for the reason above.

We do agree that there is an opportunity for clarification within the Figure caption such that readers can understand the whole thought process behind the reason behind the equation. We added the following explanation in the caption for Figure 1(b):

“Common CEA impact proposition and the respective quantifiable climate mitigation potential are shown in purple and beige. Potential global warming potential benefits from common CEA impact proposition can be respectively quantified as the first step. The MET is a measure of operational prudence calculated by dividing the quantifiable climate mitigation potential with the respective electricity emission factor.”

➤ **Comment:**

Page 3, Line 63 – Amend *“influence”* to *“influences”*

Response:

Thank you, done.

➤ **Comment:**

Page 3, Line 71 – Amend “*use-productivity*” to “.”

Response:

We thank the reviewer for pointing out the need to maintain better consistency in the terminology suggested and its abbreviation. We have amended the MET as the abbreviation for Maximum Energy-use Threshold, instead of Maximum Energy use-productivity Threshold. The application of this threshold on a basis of per kilogram of crop was highlighted in throughout the text.

➤ **Comment:**

Page 3, Line 88: “*In terms of paper writing and presentation of scientific conducted work it would be preferable to include a separate chapter concerning the methodology that was followed for data collection. the system boundaries of your research, the data collection and the examined scenarios, possible hypotheses and cases that were applied*”

Response:

We thank the reviewer for highlighting the need to clarify on the methodology within the writing.

We have detailed our methods separately in the “Methods” section of the article. We believe this suggestion might be linked to the reviewer’s later comment at Page 8, Line 285 where the reviewer commented: “*why method is after the Conclusions? It should be under the Introduction and before Results*”.

Hence, we believe this suggestion might have been raised due to the order of reading the separate chapters and does not necessitate a new methodology chapter. With regards to the order “Methods” being after “Conclusions”, we have chosen to keep the article format consistent with Nature Communications Section order.

We welcome further clarification on this suggestion, if we have misunderstood the suggested amendments required.

➤ **Comment:**

Page 3, Line 92 to Line 103 – *“all these information on data selection and case scenario development belong to a Materials and Method chapter”*

Response:

We thank the reviewer for suggesting the highlighted paragraph to be reordered for better structure and reading flow to the audience.

The original intent of the paragraph was to inform the audience the selection of crops in our results early and why we did so. We can understand the reviewer’s perspective on how the paragraph can potentially be distracting before the results and figures were introduced and hence could be shifted to the materials and methods chapter.

We have now repositioned the explanation after introducing the first result so that it explains the selection of crop within the figure. We have also swapped the order of Panel A and B, such that the audience’s first focus will be on our new results, and the relevance of the result in context of current energy use performance from literature will come after.

➤ **Comment:**

Page 4, Figure 2, Panel B – *“the MET calculated in this figure is under which CEA system? MET can significantly vary depending on the equipment used for production, the productivity rate etc. Is it uniform for every crop species presented? please specify.”*

Response:

We thank the reviewer for the clarification and agree that the relevance for MET to which form of CEA can be better clarified.

Because the MET is calculated based on external contextual factors such as emissions from current supply chain (the incumbent option) and local electricity grid, the applicability of the MET is not constrained to a specific form of CEA and does not vary depending on the equipment used.

We have added the above explanation, verbatim, in the introduction of MET on Page 3, 2nd Paragraph.

On a practical level, the different forms of CEA will find the challenge of meeting the target differently. A greenhouse operator will find meeting the MET to be easier than a plant factory operator. The MET being technology agnostic is an attractive proposition to policy makers because they form a minimum basis for CEA industry to rally towards.

We have added the above explanation, verbatim, in the final paragraph of Discussion, Page 10, last Paragraph.

The MET does vary for different crop because the existing trade options will change, and hence we presented separate MET for separate crops.

➤ **Comment:**

Page 5, Line 162 – *“please explain the abbreviation”*

Response:

We thank for reviewer for highlighting the importance of explaining the abbreviation, “SSP2-Base”.

We agree and have done so in the original sentence containing that abbreviation. The original sentence is: *“Here, with a shared socioeconomic pathway (SSP) extrapolated from historical patterns and a target equivalent to 6.0 W/m², the scenario “SSP2-Base” was assumed”*.

We understand that the phrasing of the explanation can be improved and have restructured it to:

“Here, the scenario “SSP2-Base” was assumed, where the shared socioeconomic pathway (SSP) was extrapolated from historical patterns and possesses a base emission target causing a change in energy balance equivalent to 6.0 W/m²”.

The explanation and reference to SSP2-Base was reiterated again in the Methods chapter.

➤ **Comment:**

Page 7, Line 236 – *“Add citation please”* for the term FAO reporting.

Response:

Consistent with prior section where a citation was added to reference the FAO database where reporting nations importing other produce can be found, we have added the same citation here to point towards the right resource.

We thank the reviewer for the keen observation.

➤ **Comment:**

Page 8, Line 270 – Amend *“all that is produced”* to *“the total production”* .

Response:

Thank you, done.

➤ **Comment:**

Page 8, Line 285 - *"why method is after the Conclusions? It should be under the Introduction and before Results"*

Response:

As explained in response to reviewer's previous comment at Pg 3, line 88, we have positioned the "Methods" section after "Conclusion" to be consistent with Nature Communications article format.

➤ **Comment:**

Page 9, Line 302 - *"please number the equations"*

Response:

We thank the reviewer for the keen observation. We have numbered all the equations here.

Response to comments by Reviewer 2

➤ Introductory Comment:

First, thanks to the authors and the editor for the opportunity to review this manuscript. It was interesting and presented a novel way of thinking about spatio-temporal optimization for CEA. As I understand it, the manuscript aims to explore the tradeoffs between CEA energy use (and associated carbon footprint) and the limitations of traditional ag, including long-distance travel and expanding land use.

Overall, I think this is a useful study for the Nature Comms audience, since it encourages folks to think spatially about this particular environmental challenge, and it offers general guidance on the types of situations that might justify extensive investment in CEA. “

“That said, I do have some comments that I think are important to address before the manuscript is ready to see publication. Overall, I think the article overstates the conclusions that can be drawn from these methods. This is not a test of low-carbon CEA – that would demand an LCA or other methods that includes inputs and infrastructure.

Response:

The reviewer is right and that the LCA methodology is the appropriate tool for quantifying detailed environmental impacts. We are keenly aware of the key differences between our approach and LCA, and we have differentiated the two approaches in our original manuscript.

More detailed clarifications and amendments in the manuscript are elaborated below in response to the specifics. We thank the reviewers for highlighting the need to be cautious about the framing of the results.

➤ Introductory Comment:

I also think the human dimensions of the proposed interventions are largely overlooked and should be considered with more care. Finally, I think the figures and phrasing of some of the methods are difficult to interpret, and the authors should endeavour to make clearer connections between the actual numbers they calculated and the conclusions they draw. I would be happy to re-review it once these changes have been made.

I list specific versions of these comments in no particular order for consideration:

Response:

We thank the reviewer for highlighting the need to consider human dimensions of CEA concerning significant import and agricultural land use. Indeed, human and social dimension are an important aspect of a holistic assessment, and opportunities exists for

future work using holistic methodologies such as food sovereignty framework¹⁻³. We have also improved our figures to make interpretation easier.

More detailed clarifications and amendments in the manuscript are elaborated below in response to the specifics.

➤ **Itemised Comment:**

1. “The calculated threshold is fundamentally a balance between anticipated CEA carbon footprint solely from energy use and avoided carbon footprint implicit in CEA impact propositions (Fig. 1b).” - I am concerned about the framing of CEA as environmentally friendly prior to any LCA work.

As the authors themselves point out, there are a number of infrastructural investments that must be amortized as carbon per kilogram product – and other recent work in this family of journals has shown that infrastructure is perhaps the most important factor in LCA of urban ag (Hawes et al., 2024).

There were similar findings in other, older CEA work (Goldstein et al., 2017). It might be more accurate to say that the operation of CEA could mitigate trade-offs under confluent conditions, but I don’t think that the authors demonstrate clearly that CEA is low-carbon even if its operation is.

This is ultimately a phrasing and framing issue, I certainly don’t expect the authors to conduct a full-blown LCA to address this.

Hawes, Jason K., Benjamin P. Goldstein, Joshua P. Newell, Erica Dorr, Silvio Caputo, Runrid Fox-Kämper, Baptiste Grard, et al. “Comparing the Carbon Footprints of Urban and Conventional Agriculture.” *Nature Cities*, January 22, 2024, 1–10. <https://doi.org/10.1038/s44284-023-00023-3>.

Goldstein, Benjamin, Michael Hauschild, John Fernández, and Morten Birkved. “Testing the Environmental Performance of Urban Agriculture as a Food Supply in Northern Climates.” *Journal of Cleaner Production* 135 (November 2016): 984–94. <https://doi.org/10.1016/j.jclepro.2016.07.004>.

Response:

We have amended the manuscript to make clear to the readers at multiple points the need for a further LCA, especially for certain forms of CEA where infrastructure can be the primary contributor of emissions.

Specifically:

(a) The need for LCA is highlighted at –

- Page 3, Paragraph 2 – *“In this work, we developed and calculated a Maximum Energy-use Threshold, or MET, per kilogram of crop, for different countries and different future scenarios. This helps us in identifying at an aggregate level, favourable scenarios which have the potential to realise the climate mitigation potential of CEA. For CEA ventures operating within this measure of operational prudence, more detailed LCA evaluating the full environmental impact should then be embarked on.”*
- Page 3, Paragraph 3 – *“In doing so, the aim of this article is to determine the confluence of conditions that could potentially allow CEA to mitigate Energy-Food-Climate Trade-off, before heavy investment of time and resources into an LCA.”*

(b) Especially for forms of CEA where infrastructure can be key (including citation suggested by reviewer) –

Page 9, Paragraph 1 of Discussion – *“Here, the Maximum Energy-use-Threshold (MET) is introduced as a resource use metric balancing energy-food-climate consideration. It is a tool complementary to environmental impact assessment methodology such as LCA, where LCA could include impacts from other sources, such as infrastructure. LCA remains important as some form and scale of CEA might require little energy but with most of its emissions stemming from infrastructure – such as a small-scale urban farm^{4,5}. Here, because energy use is a key carbon emissions contributor in larger scale CEA, the MET provides a measure of prudence where operation exceeding it will not position CEA to achieve a lower carbon food system, even without conducting an LCA. Operating below this measure of prudence places a CEA scenario in good standing for a more detailed analysis using LCA to confirm its full environmental impact profile.”*

➤ **Itemised Comment:**

2. The manuscript should be edited once more to remove odd words or grammatical tricks here and there, like an extra “be” at line 77. “Transiting” at line 156. Sentence at 247. Etc

Response:

Thank you, done.

➤ **Itemised Comment:**

3. In general, the figures could use some work. Aesthetically, they are quite nice, and they communicate a simple message quite well. But they fail to communicate the key message of the results section, which is where and under what circumstances the EUP exceeds or is below the MET. I don’t know that I think the current figures need removed or changed, but an additional panel that shows the EUP divided by MET relative to 1 might be helpful.

Response:

For better clarity, we have split the two panels of the original Figure 2 and added a new feature to the original panel of Figure 2b, which is now presented in the article as Figure 3. Specifically, we have enhanced all the figures in the following manner:

- a) Page 4, Figure 2: MET considering combinations of contextual parameters –
 - The average energy use-productivity for indoor farming and greenhouses were extracted from literatures and illustrated in the panel as a line marker. Countries with MET higher than this average denotes a favourable scenario. We have clarified this new addition in the text (Page 4, Paragraph 1) as well: *“The average EUP for different crops and form of CEA was extracted from literatures to visualise how current CEA compare to the METs. Countries with MET above these markers denote a favourable scenario for CEA implementation today, given the current operational state of CEA (GH – Greenhouse, or IF – Indoor Farming).“*

- b) Figure 3: EUP estimates from literature and energy prudence index–
 - An “Energy Prudence Index” was added where a positive displacement to the right of the line denotes higher energy use than the MET and that literature would not be sustainable. A negative displacement to the left of

the line denotes lower energy use than the MET and hence further time and resources should be invested in a LCA to study the set-up in that literature source.

We have clarified this new addition in the text (Page 6, Paragraph 1): *“To evaluate the climate mitigation potential of CEA in particular country, a “Energy Prudence Index” is suggested here. This represents the ratio between the difference in reported energy use and MET. A positive value in the Energy Prudence Index signifies a higher energy use than the MET, and a negative value in the index signifies a lower energy use than the MET.”*

- c) Page 7, Figure 4a and Figure 4b; Page 8, Figure 5a and Figure 5b – The average energy use extracted from current literatures are added into the figures. Similar to Figure 2a, this will help the reader understand that if the current average is already higher than the METs, it is unlikely those countries will provide a favourable scenario.

➤ **Itemised Comments:**

4. At line 135, there is discussion of the low carbon impacts of electricity in certain developing countries. I have to wonder about the externalities associated with siting high energy-use crop growing systems in places where the grid may be unstable or unprepared for a new, large user. While I can appreciate the idea, it seems a bit short-sighted to me.

5. Related to this, I do not understand this statement – why is Ethiopia better than the others named? -- “Resonating with a previous study that assessed macro indicators for indoor vertical farming’s feasibility and sustainability in Africa, Ethiopia stands out as the African nation suitable for CEA.

Response:

We thank the reviewer for highlighting the externalities that prospective CEA operators or planner should take note of. We fully agree that beyond the consideration of renewable energy availability and current agrifood supply chain, there will be non-ecological related factors.

While we cannot confirm specific grid development that will resolve stability and capacity issues definitively in the future, we believe the mentioned developing countries can accommodate CEA because large energy users that cannot tolerate grid instability already exists at the mentioned developing countries. Specifically, data centers. For example, in Ethiopia, Raxio, a Tier III certified data centre with less than 1.6 hours downtime per year has been operating since November 2022⁶. In Congo, the National Data Center is anticipated to be inaugurated in end of 2025⁷. With regards to the availability of energy resources to accommodate new users, for countries such as Ethiopia, renewable energy resources remain largely unexploited⁸.

In a separate study, development indicators of African nations were extracted from World Bank as well as FAO and synthesized into a feasibility index. The index include development indicators in macro areas such as food security, climate change vulnerability, infrastructure and others. In that work, Ethiopia has a synthetic feasibility index that is favourable for indoor vertical farming with artificial lighting in Africa⁹. On the other hand, for Congo and Democratic Republic of Congo, they have been classified as “not favourable” when considering the fourteen macro-indicators. Ethiopia scored seven favourable classifications while Congo and Democratic Republic of Congo scored four favourable classifications each.

In conjunction with our results presented here, where Ethiopia has one of the cleanest electricity grids in Africa and thus allowing a higher MET, there is high confidence that Ethiopia is a standout African nation for CEA.

We thank the reviewer for highlight this opportunity to clarify this statement. We have now added this clarification in Page 5, Paragraph 1 – “*Resonating with the results here, another*

study where indoor vertical farming feasibility scores are synthesized from development indicators, Ethiopia scored favourably to host indoor vertical farming in macro indicators such as food security, climate change vulnerability, infrastructure, and others⁹. Though externalities such as grid stability and capacity can be a concern in developing countries such as Ethiopia, siting CEA would be possible as energy intensive, technology infrastructure like data centres already exists and provides a template for planning high energy use, high cooling load ventures⁶.”

➤ **Itemised Comment:**

6. I am very unclear on what type of food is being offset in the agricultural land substitution scenario. Is this local food production replaced by CEA? Or imported food? If imported food, the explanation doesn't seem to make sense to me. The conclusion seems to indicate that this section is focused on how exporting countries might reduce their footprint – this is interesting and perhaps more internally consistent, but I have some concerns as discussed in my next point.

Response:

We thank the reviewer for highlighting this point and providing the opportunity to clarify. In that section, the carbon opportunity cost from using currently occupied land to produce food locally was used to calculate the MET. Hence, it is taken to substitute local food production and with the assumption that the agricultural land is converted to provide environmental services (carbon sequestration). We acknowledge that the short mention of “*wheat and soybean producing nations.*” at Page 7, Line 236 of the original manuscript can be elaborated to eliminate any confusion.

The paragraph introducing the results has been expanded to make clear of our assumptions and what the result implies. It now reads at Page 8, Paragraph 3:

“Here, a carbon opportunity cost was valued for land currently occupied to produce wheat and soybean in order to account for this opportunity of carbon sequestration¹⁰. The countries analysed include wheat and soybean producing nations as reported to FAO¹¹. The result in Figure 4 shows the MET of wheat and soybean CEA if the CEA production now substitutes agricultural land outputs and the agricultural land can now be restored to provide environmental services.”

➤ **Itemised Comment:**

7. Overall, I remain a bit unconvinced of the idea that CEA inherently reduces the land use for agriculture in other locations. These sorts of telecoupling effects are unpredictable at best, and things like local labor markets are unlikely to simply vanish if the demand from a particular export partner is reduced. For that matter, the authors still need to talk about the in-migration to cities that would be required, the upskilling of agricultural labor, etc. If CEA requires less employees per kg of produce, it seems unlikely that the land use transition would be dramatic, since the original farmers still need a source of income.

Response:

We thank the reviewer for highlighting the social and economic dimensions that have to be considered for CEA to be a successful in transforming the agrifood system.

CEA inherently increases the yield output (kilogram crop per unit of land area) on an arable land or non-arable land in an urbanised environment. Hence, the ability of CEA to reduce land use is not dissimilar to investigations of the link between agricultural intensification and land-sparing¹². As exploitable gap between farm yields and genetic yields potential closes¹³, CEA provides a form of precision agriculture that can increase yield output, but without land constraint as more storeys can be built upon.

Globally, there are observations supporting the general trend of increasing yield output with slowing land use, and more land will be needed without agricultural intensification¹⁴. However, the link of agriculture intensification and land sparing remains highly context specific^{15,16,17}. This lends to the highly context dependent nature of CEA to mitigate energy-food-climate trade-off. Here, a focus on balancing energy-food-climate trade off was placed because of the urgency to balance the demand and impact on natural resources by agrifood systems. Nonetheless, we agree that the social-economic conditions should be elaborated in the discussion to prompt readers to think beyond the scope of this work and also provide opportunities for future work.

Page 9, Paragraph 2 of Discussion reads:

“It is worth highlighting that a particular CEA system operating below the MET does not confirm the accomplishment of CEA impact proposition such as international import elimination or agricultural land restoration. These agrifood system transformation will require not just technical innovation in CEA operation, but also confluent social-economical and policy conditions involving diverse stakeholders¹⁸. Particularly for the link between agricultural intensification through CEA and land sparing, stimulative conditions are required to promote land sparing for environmental services¹⁵⁻¹⁷. They include policies for conservation-set aside programs with financial instruments to remunerate landowner for the alternative use of rural lands^{19,20}, and potentially increasing

grain imports at low commodity price¹². The holistic evaluation of sustainability encompassing ecological, social and economic aspect would require other conceptual frameworks¹⁻³, and presents opportunities for future work. The MET provides a measure of operational prudence in order to truly realise the climate mitigation benefit of these sustainability propositions. Here, there are a few applications in which the MET can already be valuable.”

➤ **Itemised Comment:**

8. The authors talk about an expanded technical lifespan for solar, but I don't think they ever say what 50 years is an alternative to. What is the technical lifespan in EcolInvent? This should be in the main text (I see it in the methods, but this is a Nature Comms piece, not all readers will keep going to find the methods).

Response:

The technical lifespan of a solar panel is 30 years. This is now included in Page 9, Paragraph 2 where it reads:

“To calculate the emission factor of a 50-year technical life span solar panel, the existing life cycle inventory of a 30-year technical life span solar panel was extended to continue at a degrading annual yield for another 20 years.”

We thank for reviewer for highlighting the need for this information to be presented earlier.

➤ **Itemised Comment:**

9. I'm a bit confused in the methods section with Eqn. 1. If the country is land-locked, all food is assumed to be ground-transported? This doesn't make sense for, for example, import of bananas from Colombia to Switzerland. Some of that distance is, of course, by sea.

Response:

We thank the reviewer for the opportunity to clarify in this detail.

Eqn.1 relates to *pairs* of importing and exporting nation, and only when both the countries are land-locked, will the food be assumed to be ground transported. For example, while Switzerland is a land-locked country, only trading between another land-locked country such as Austria would use ground-transportation. Colombia is not a land-locked country; hence it will involve maritime transport.

We have taken the opportunity to clarify this in the methods section (Page 11, Paragraph 2), and the equation itself:

“For trading pairs where both countries are land-locked – without access to ports and directly accessible to each other overland – the shortest road distance between capitals was used instead”

➤ **Itemised Comment:**

10. I appreciate that the authors try to differentiate between cooled lorry use and dry goods, but does Table 1 in the methods suggest that the carbon footprint of a refrigerated lorry is lower than that of an unspecified lorry? This might indicate that the “unspecified” option includes some combination of a variety of lorries, which wouldn’t really be achieving the authors’ goals.

Response:

We thank the reviewer for the opportunity to clarify in this detail.

Based on Ecoinvent 3.9.1 datasets, the impact score/ carbon footprint of a refrigerated lorry is indeed lower than that of an unspecified lorry. Here, the “unspecified” option refers to an aggregate of different lorry class (metric ton) and EURO emission standard. While the distribution of EURO emission standard lorry for both datasets are similar, the small difference in assumed lorry class (3.5-7.5, 7.5-16, 16-32, >32 metric ton) contributed to a higher impact score for unrefrigerated lorry dataset. In the refrigerated lorry dataset, only 16-32 and >32 metric ton lorry class was considered. However, in the unrefrigerated lorry dataset, smaller lorry class were included. As compared to >32 metric ton lorry class, smaller lorry class can have up to 5.31 times the emissions.

We selected to use an unspecified option as we do not have granular information about the specific lorry/EURO class used in a specific leg of a logistic chain. While it is not ideal, we have opted to use this instead of assuming a specific class for countries in a specific region.

➤ **Itemised Comment:**

11. The authors should be more specific about the programming languages used.

Response:

We have rephrased the explanation under “Code Availability”:

“Computational steps are described in the Methods, as well as in original works referenced in this study. Scripts performing the computation are written in Python programming language and available from the corresponding author upon reasonable request. The custom code that support the findings of this study are also available on Code Ocean (10.24433/CO.2717777).”

We thank the reviewer for the opportunity to clarify in this detail.

➤ **Itemised Comment:**

I have not had time to replicate the code, but assume that the authors were careful and thorough.

Response:

We understand that the reviewer may not have time to replicate the code. Nonetheless, we are appreciative of the reviewer’s comment and time in reviewing the manuscript. Thank you again.

References

1. Wittman, H. Food sovereignty: An inclusive model for feeding the world and cooling the planet. *One Earth* **6**, 474–478 (2023).
2. Oteros-Rozas, E., Ruiz-Almeida, A., Aguado, M., González, J. A. & Rivera-Ferre, M. G. A social–ecological analysis of the global agrifood system. *Proceedings of the National Academy of Sciences* **116**, 26465–26473 (2019).
3. Ruiz-Almeida, A. & Rivera-Ferre, M. G. Internationally-based indicators to measure Agri-food systems sustainability using food sovereignty as a conceptual framework. *Food Secur* **11**, 1321–1337 (2019).
4. Hawes, J. K. *et al.* Comparing the carbon footprints of urban and conventional agriculture. *Nature Cities* **1**, 164–173 (2024).
5. Goldstein, B., Hauschild, M., Fernández, J. & Birkved, M. Testing the environmental performance of urban agriculture as a food supply in northern climates. *J Clean Prod* **135**, 984–994 (2016).
6. Raxio Group. Raxio Continues Support of Africa’s Digital Transformation and Digital Economy with Launch of New Flagship Data Centre in Ethiopia. <https://www.raxiogroup.com/raxio-continues-support-of-africas-digital-transformation-and-digital-economy-with-launch-of-new-flagship-data-centre-in-ethiopia/> <https://www.raxiogroup.com/raxio-continues-support-of-africas-digital-transformation-and-digital-economy-with-launch-of-new-flagship-data-centre-in-ethiopia/> (2023).
7. African Development Bank. Congo: New data centre funded by African Development Bank will cement national and subregional digital sovereignty. <https://www.afdb.org/en/news-and-events/congo-new-data-centre-funded-african-development-bank-will-cement-national-and-subregional-digital-sovereignty-70847> <https://www.afdb.org/en/news-and-events/congo-new-data-centre-funded-african-development-bank-will-cement-national-and-subregional-digital-sovereignty-70847> (2024).
8. Yalew, A. W. The Ethiopian energy sector and its implications for the SDGs and modeling. *Renewable and Sustainable Energy Transition* **2**, 100018 (2022).
9. Paucek, I. *et al.* A methodological tool for sustainability and feasibility assessment of indoor vertical farming with artificial lighting in Africa. *Sci Rep* **13**, 2109 (2023).
10. Searchinger, T. D., Wiersenius, S., Beringer, T. & Dumas, P. Assessing the efficiency of changes in land use for mitigating climate change. *Nature* **564**, 249–253 (2018).

11. Food and Agriculture Organization. Food and Agriculture Organization Corporate Statistical Database. <https://www.fao.org/faostat/en/#data/TM> (2022).
12. Rudel, T. K. *et al.* Agricultural intensification and changes in cultivated areas, 1970–2005. *Proceedings of the National Academy of Sciences* **106**, 20675–20680 (2009).
13. Cassman, K. G. Ecological intensification of cereal production systems: Yield potential, soil quality, and precision agriculture. *Proceedings of the National Academy of Sciences* **96**, 5952–5959 (1999).
14. Hannah Ritchie. Yields vs. land use: how the Green Revolution enabled us to feed a growing population. *Our World in Data* (2017).
15. Pratzler, M. *et al.* Agricultural intensification, Indigenous stewardship and land sparing in tropical dry forests. *Nat Sustain* **6**, 671–682 (2023).
16. Lin, M. & Huang, Q. Exploring the relationship between agricultural intensification and changes in cropland areas in the US. *Agric Ecosyst Environ* **274**, 33–40 (2019).
17. Ceddia, M. G., Bardsley, N. O., Gomez-y-Paloma, S. & Sedlacek, S. Governance, agricultural intensification, and land sparing in tropical South America. *Proceedings of the National Academy of Sciences* **111**, 7242–7247 (2014).
18. Barrett, C. B. *et al.* Bundling innovations to transform agri-food systems. *Nat Sustain* **3**, 974–976 (2020).
19. Barrett, C. B. Overcoming Global Food Security Challenges through Science and Solidarity. *Am J Agric Econ* **103**, 422–447 (2021).
20. Xu, Z. *et al.* Grain for Green versus Grain: Conflict between Food Security and Conservation Set-Aside in China. *World Dev* **34**, 130–148 (2006).

General Response To Both Reviewers

We would like to thank both reviewers again for their valuable comments and for providing us an opportunity to enhance the clarity of this work.

Specifically, we thank Reviewer 1 for highlighting the need to enhance clarity in the methodological assumptions as well as adding relevant discussion issues. The amended manuscript has benefited greatly from these suggestions, with significant additions to the discussion and two additional figures. One as a visual flowchart to help readers understand the method better, and the other to illustrate an example of how MET could be institutionalised.

We also thank Reviewer 2 for the careful reading, spotting minor grammatical corrections in the manuscript and minor annotation correction in one of the figures. Adding to the improvements from previous round of revision, the quality of this article has benefited significantly from these suggestions.

For ease of perusal, specific responses to Reviewer 1 can be found from HERE, and specific responses to Reviewer 2 can be found from HERE.

We thank both reviewers again for their insightful comments and welcome any further clarification.

Point-by-point response to the reviewers' comments

Response to comments by Reviewer 1

➤ Comment:

The MET concept is novel, but its relationship to existing environmental performance metrics (e.g., specific LCA indicators, energy use productivity benchmarks) could be more explicitly articulated. How does MET add actionable value beyond existing LCA thresholds?

Response:

We thank the reviewer for the opportunity to clarify the value of MET.

The MET adds value by allowing operators to act on improving their operational prudence towards an objective measure and allow policymakers to objectively form policies that could incentivise operational prudence. Elaborating on the reason for why LCA results and industry disclosed performance is unsuitable as an objective benchmark:

First, as LCA results are highly specific to a scenario, the associated energy use-productivity cannot be used as a threshold for other scenarios. Even when a scenario could have been a lower carbon option in its comparison, it does not translate to another context. Hence, the need for a threshold complementary to highly granular LCAs.

Next, current energy use-productivity based on disclosed operations cannot be used as a benchmark because it might not be fully transparent in its internal methodology. More importantly, an industry average based on disclosed operations cannot be used because their relative performance is not grounded in a quantitative evaluation considering environmental impact parity to local, existing options.

Here, the MET has the advantage of being objective in its calculation independent of CEA operator's potential bias and is transparent in its data as well as approach. In the first draft, two ways in which the MET could be valuable was elaborated. In this revision, line 305 to 310, we highlighted the value of an objective benchmark before elaborating them. Reproduced verbatim:

“An objective and transparent benchmark is necessary to guide the development of CEA. Objective in being independent of a CEA operator's potential bias during its calculation and being transparent in its data and approach. The MET adds value here by informing operators of an objective benchmark to optimise towards and for policymakers to objectively form policies that incentivise sustainable operation. Elaborating further, here are two ways in which the MET can already be valuable.”

➤ **Comment:**

The assumption that CEA systems meeting MET will inherently achieve climate benefits seems optimistic. A discussion on potential rebound effects or partial achievement of impact propositions would be helpful.

Response:

We thank the reviewer for the suggestion to discuss on potential rebound effects or partial achievement of impact propositions. Combining with a paragraph in the discussion where other supporting social-economic and policy conditions have to be present for CEA to achieve climate benefits, we have restructured the discussion into three main sections. In the last section, a further discussion on the potential rebound effects is now included. The section (*with italic font being new explanation*) reads:

“ ***Nuances of sustainability assessment of CEA and emerging technologies.***

It is worth highlighting that a particular CEA system operating below the MET does not confirm the accomplishment of CEA impact proposition such as international import elimination or agricultural land restoration. These agrifood system transformation will require not just technical innovation in CEA operation, but also supporting social-economical and policy conditions involving diverse stakeholders¹. Particularly for the link between agricultural intensification through CEA and land sparing, stimulative conditions are required to promote land sparing for environmental services²⁻⁴. They include policies for conservation-set aside programs with financial instruments to remunerate landowner for the alternative use of rural lands⁵⁻⁷. The holistic evaluation of sustainability encompassing ecological, social and economic aspect would require other conceptual frameworks⁸⁻¹⁰, and presents opportunities for future work.

The sustainability assessment of CEA, as well as other emerging technologies, are highly nuanced and can often cause unintended rebound consequences. For example, as CEA reduces agricultural land use, there can be more arable land available, given the assumption that stimulative conditions for land sparing are present. These stimulative conditions will include financial incentives that can be more attractive than the gains from using the land for food production. This can cause an increase in cost of goods for many sectors, threatening the promise of enhancing food security.

Alternatively, if CEA were to be hosted on non-arable land, social rebound effects can emerge. For example, to reduce CEA thermal cooling load and the cost of operation, CEA operators favours locations with cool, temperate climates and low electricity costs. Often, these places would already have communities settled there. A parallel to this rebound effect can be drawn from the dense siting of data centres in North Virginia, USA¹¹. Quality of life impact, rising local utilities bills, water scarcity, decreasing property value, declining local community support have all been reported.

The environmental consequence of substituting traditionally farmed products with CEA products demands a separate and far more complex analysis. Consequential life cycle assessment which considers the implications on marginal suppliers due to a change in supply and demand can be a first step¹². Modelling stakeholders as agents and integrating emergent behaviours from their interaction to life cycle assessment models can be the next step¹³⁻¹⁵. The environmental impact and benefits of CEA as well as emerging innovations are highly nuanced. Advanced modelling tools such as LCA will no doubt be necessary to aid decision making^{16,17}. Here, the MET provides a complementary benchmark in CEA before such detailed modelling and analysis is embarked on.”

➤ **Comments:**

Figures 2–5 are rich in data but can be challenging to interpret. Consider simplifying axis labels, highlighting key outlier countries more consistently, and integrating policy-relevant thresholds directly into visuals

Response:

We thank the reviewer for the suggestion to make the figures’ interpretation easier. With regards to simplifying the axis labels, we have chosen to retain the current version to help readers understand the specific CEA impact propositions, crop and scenarios for which the MET is calculated. On highlighting key outliers more consistently, the top five key outliers have been annotated consistently, except in cases where there are only four outliers. Lastly, with regards to integrating policy relevant thresholds directly into visuals, it resonates with the reviewer’s latter comment on how to further institutionalise MET and possible work to transition MET for policy use. This is addressed below here (under response to comment pertaining to L293).

➤ **Comment:**

For trade distance assumptions: was any validation done to assess the realism of capital-to-capital distances vs. actual supply chains?

Response:

For trade distance assumptions, Bertoli et al. calculated the capital-to-capital distances based on relevant ports (the one with the most shipping lanes) and existing maritime routes¹⁸. This approach was referenced as it provides a better estimate than a shortest straight-line estimation. The validation of the referenced work with actual supply chains is difficult because there can be many routes from more than one port between two countries, with multiple freight operators and the individual choice of freight routes not readily available.

➤ **Comment:**

How were seasonality and perishability handled in emissions calculations (especially for leafy greens and strawberries)?

Response:

As seasonality and perishability affects the supply chain choices of an importing country through fluctuating commodity prices and transportation costs, a country often have more than one trading partner for a food product. This is incorporated into the MET as it takes the weighted average trade distance considering all its trading partner in a year. The MET calculation was repeated with FAO data from 2012 to 2022 and the median of the calculations over 10 years was taken to also capture short-term supply chain changes due to unexpected events.

In addition, because leafy greens and strawberries are perishable and requires cooling during transportation, the emission factor used in the analysis of leafy greens relate to sea freight container with reefer cooling, and in the case of strawberries, the emission factor correspond to freight aircraft transport with reefer cooling.

➤ **Comment:**

L3. Confluent: Maybe the term “contextual conditions” fits better

Response:

We thank the reviewer for the suggestion to clarify the title. We now propose a more fitting title after considering the emphasis over the two revisions and novelty of MET that resonated with both reviewers. The new title for this article is:

“Contextual Conditions Define Maximum Energy-use Threshold in Low-Carbon Controlled Environment Agriculture for Agri-Food Transformation.”

➤ **Comment:**

L72: MET is an excellent construct, but its definition and derivation process could be clearer. Consider including a visual flowchart summarising its components (inputs, calculations, thresholds).

Response:

We thank the reviewer for highlighting the need to clarify on the definition and derivation process. We have separated out this definition from the original Figure 1 and now created a new Figure 2 where a visual flowchart summarising the inputs, database and brief description of the calculation steps.

➤ **Comment:**

L86-89: Have you deliberately excluded non-energy impacts in the MET (e.g., materials, embodied carbon of infrastructure)?

Response:

We thank the reviewer for the clarification. Yes, we have excluded non-energy impacts from materials and embodied carbon of infrastructure in the MET. This is because in most large-scale CEA facilities, energy use is the key emissions contributor. However, there can be scenarios where low energy is required in a farm and embodied carbon of infrastructure will be significant, for example in small-scale urban farms. This was highlighted in the discussion section, where the need for LCA remains as more emissions contributors are taken into consideration. The exclusion of materials and embodied carbon of infrastructure lends to the intent of MET being a technology agnostic measure, complementary to LCAs.

➤ **Comment:**

L106 (Fig.3): The interpretation of MET vs. actual EUP is clear in figures, but some textual explanations could be more concise and better linked to key implication

Response:

We agree and thank the reviewer for the suggestion. We have rephrased and made the related explanation from line 164 to 170 more concise. Reproduced verbatim:

“Based on literature, almost all of the current CEA setups operate at EUP several multiples of the MET. Only in setups where extremely low energy is used and where short shelf life produce such as strawberries are air freighted, their operations hold potential to be a lower carbon option. For example, utilising a hydraulic rotating greenhouse setup in Singapore for CEA of lettuce can be achieved at an EUP of 0.021 kWh/kg¹⁹, lower than the MET (Supplementary Data 1). As a land-scarce nation intensifying agri-food industry investments to bolster food security²⁰, it would be beneficial for Singapore to invest in low-energy greenhouses for lettuce cultivation.”

➤ **Comment:**

L126-128: Could the authors clarify how air freight assumptions for strawberries were verified across the sample countries?

Response:

We thank the reviewer for the clarification. This has been a challenge, as with the previous validation of the shipping route. While we cannot verify the actual air freight distances due to the lack of available data, our approach here of using the shortest straight-line distance is aligned with the Great Circle Distance approach used by the International Civil Aviation Organization's (ICAO) air freight carbon emission calculation methodology²¹. In the referenced methodology, the verification of the actual flown distance was determined to be not feasible for the time being as well. Combining with the previous clarification on shipping distance validation, we have now added a short explanation in the Methods section to highlight this challenge and assumption. Reproduced verbatim:

“Due to the unavailability of actual shipping and air freight data, validation of sea freight distance and air freight distance was not feasible here. However, the shortest straight-line approach using a geodisc here is consistent with the shortest straight-line approach using a Great Circle Distance by the International Civil Aviation Organisation in its freight carbon emission calculator²¹ “

➤ **Comment:**

L132: In the discussion of favourable locations (e.g., Ethiopia, Paraguay), have you considered local grid reliability and cold-chain logistics as potential limiting factors?

Response:

We thank the reviewer for highlighting the two potential limiting factors. For the first factor of local grid reliability, it was addressed in the previous revision, where grid stability and capacity could indeed be a concern. However, with data centres that are highly energy demanding and does not tolerate grid instability already existing in developing countries such as Ethiopia, the energy intensive technology infrastructures for data centres are a template for siting CEA. With regards to cold-chain logistics, the developing cold-chain capabilities would be a potential driving factors for CEA to be sited closely to demands or ports to maximise perishable produce shelf life.

➤ **Comment:**

L210 (Fig. 4): How sensitive are your MET results to PV efficiency improvements vs. emission factor assumptions?

Response:

In the sensitivity analysis of MET, the results were directly affected by changes in emission factors due to several technological factors. Namely, (1) full reliance on PV, (2) full reliance on PV and extended life span of PV, and (3) in projected grid mix under different SSP. PV efficiencies were improved in the third scenario using prospective energy grid emission factors but its effect not isolated from other energy grid developments. There, PV efficiency were assumed in *premise*²² to increase at different time points:

% module efficiency	micro-Si	single-Si	multi-Si	CIGS	CIS	CdTe
2010	10	15.1	14	11	11	10
2020	11.9	17.9	16.8	14	14	16.8
2050	12.5	26.7	24.4	23.4	23.4	21

While a separate sensitivity analysis of only PV efficiency improvements was not conducted, parallels can be drawn from sensitivity scenario two, where emission factors were reduced due to increased power generation over the course of a longer service life (148,575.72 kWh over 50 years). Initially, the power generated over a 30-year PV lifespan was calculated to be 90,997.2 kWh at ~15.1% efficiency. Assuming a 76% efficiency increase to 26.7% panel efficiency, the total power generated over a 30-year lifespan would be around 160,115.1 kWh. This is a ~7% increase in the total power generated over a 50-year lifespan PV panel with lower panel efficiency. Consequently, the MET would increase by the same percentage.

Hence, we could assume the MET is sensitive to PV efficiency increase, as it is to PV technical lifespan increase, both increasing the total power generated by a PV unit.

➤ **Comment:**

L293: In the discussion of favourable locations (e.g., Ethiopia, Paraguay), have you considered local grid reliability and cold-chain logistics as potential limiting factors? Other interesting issues that can be explored in the discussion section are: a) Can the authors reflect more on how MET could be institutionalised — e.g., integrated into CEA certification, subsidies, or siting decisions? b) What would be required to transition MET from research to policy use — what validation or consensus building is needed?

Response:

We thank the reviewer for suggesting additional issues that could be of interest in the discussion. We have added a new subsection in the discussion. Reproduced verbatim for easy reference:

“ **Next Steps for MET**

Beyond this first calculation of MET for different countries, more granular data will be needed to quantify MET more precisely and avail more specific policy use cases. Validation with national, state, or province-level trade data obtained through local agencies could be feasible, instead of relying on aggregated data, thereby producing a more precise quantification of the MET at a regional level. While MET cannot be used to certify the sustainability of a CEA operation, policymakers can use MET to promote sustainable practice of CEA.

Figure 1 Possible ways to institutionalise MET to encourage sustainable controlled environment agriculture. The two-colour banners denote operational range where MET or LCA is more relevant. The asterisk symbol denotes a percentage that policymakers can adjust or add more increments in between. Some of the suggested policy levers are listed below each milestone.

Figure 7 illustrates one possible way to do this. For operations above the MET, where large-scale CEA is highly unlikely to be sustainable even without performing an LCA, such unsustainable practices should be limited in production volume, allowing CEA operators to scale up in production volume only upon demonstrating an improvement. Incentives can come in the form of co-funded research grants, one off sustainability practice grants, or subsidies to perform full LCA. Only when operators are below the MET and certified through an LCA that it is a lower carbon option, preferential electricity rate or access to carbon trading as a seller can then be made available. Consensus building with operators will be critical in determining parameters such as the appropriate MET milestones and production volume limitations to not deter new entrants or stifle small start-ups and instead motivate sustainable innovation practices. Here, the approach and formulation to calculate a MET based on contextual conditions paves the way forward for such strategies.”

Response to comments by Reviewer 2

➤ Comment:

I think the authors need to make a bit clearer in the main text what is considered in the MET. Specifically, I'd add a few sentences in the introduction or early results explaining how the carbon emissions per imported produce were found.

Response:

We thank the reviewer for highlighting the need to clarify what is included in the MET. This resonates with a previous comment by Reviewer 1 as well. We have now included a new figure (Figure 2) as a visual aid summarising the inputs and brief description of the calculation steps. This will allow readers to understand the MET conceptualisation and calculation without necessarily referring to the Methods.

➤ Comment:

At line 77, "The calculation" should be clarified. Which calculation? MET?

Response:

We thank the reviewer for the clarification and have added "MET" in line 77 to specify that it means "MET calculation".

➤ Comment:

A number of places need another grammar check. For example, "resource use are" at line 58 and "the" is missing in "within local food system" at line 85. There are other examples, just needs a careful copy-edit

Response:

We thank the reviewer for the careful reading. Indeed, we have also spotted a number of places that require grammatical corrections. We have now corrected the grammar within the entire manuscript to the best of our knowledge.

➤ **Comment:**

I think Fig 4 is missing some of the orange annotations that indicate where IF and GH fall on the MET spectrum.

Response:

We thank the reviewer for their sharp observation. We have amended the figure. In Figure 4, the MET for Strawberry does not have an orange annotation as it does not have an average estimate for indoor farming based on current literature (as noted in the figure caption). However, the reviewer is right that we have missed out two orange arrow annotation for Wheat and Soybean, denoting that from current literatures, average estimates are not capture within the figure's displayed range. We have checked the other figures, and they are correct. We thank the reviewer for highlighting this.

References

1. Barrett, C. B. *et al.* Bundling innovations to transform agri-food systems. *Nat Sustain* **3**, 974–976 (2020).
2. Ceddia, M. G., Bardsley, N. O., Gomez-y-Paloma, S. & Sedlacek, S. Governance, agricultural intensification, and land sparing in tropical South America. *Proceedings of the National Academy of Sciences* **111**, 7242–7247 (2014).
3. Lin, M. & Huang, Q. Exploring the relationship between agricultural intensification and changes in cropland areas in the US. *Agric Ecosyst Environ* **274**, 33–40 (2019).
4. Pratzler, M. *et al.* Agricultural intensification, Indigenous stewardship and land sparing in tropical dry forests. *Nat Sustain* **6**, 671–682 (2023).
5. Barrett, C. B. Overcoming Global Food Security Challenges through Science and Solidarity. *Am J Agric Econ* **103**, 422–447 (2021).
6. Xu, Z. *et al.* Grain for Green versus Grain: Conflict between Food Security and Conservation Set-Aside in China. *World Dev* **34**, 130–148 (2006).
7. Rudel, T. K. *et al.* Agricultural intensification and changes in cultivated areas, 1970–2005. *Proceedings of the National Academy of Sciences* **106**, 20675–20680 (2009).
8. Oteros-Rozas, E., Ruiz-Almeida, A., Aguado, M., González, J. A. & Rivera-Ferre, M. G. A social–ecological analysis of the global agrifood system. *Proceedings of the National Academy of Sciences* **116**, 26465–26473 (2019).
9. Ruiz-Almeida, A. & Rivera-Ferre, M. G. Internationally-based indicators to measure Agri-food systems sustainability using food sovereignty as a conceptual framework. *Food Secur* **11**, 1321–1337 (2019).
10. Wittman, H. Food sovereignty: An inclusive model for feeding the world and cooling the planet. *One Earth* **6**, 474–478 (2023).
11. Robert Leslie *et al.* Inside the secretive world of America’s AI data centers. *Business Insider* <https://www.businessinsider.com/inside-the-secretive-world-of-americas-ai-data-centers-2025-9> (2025).
12. Schaubroeck, T. Relevance of attributional and consequential life cycle assessment for society and decision support. *Frontiers in Sustainability* **Volume 4-2023**, (2023).
13. Hicks, A. Seeing the people in LCA: Agent based models as one possibility. *Resources, Conservation & Recycling Advances* **15**, 200091 (2022).

14. Larrea-Gallegos, G., Marvuglia, A., Navarrete Gutiérrez, T. & Benetto, E. A computational framework for modeling socio-technical agents in the life-cycle sustainability assessment of supply networks. *Sustain Prod Consum* **46**, 641–654 (2024).
15. Lan, K. & Yao, Y. Integrating Life Cycle Assessment and Agent-Based Modeling: A Dynamic Modeling Framework for Sustainable Agricultural Systems. *J Clean Prod* **238**, 117853 (2019).
16. Bluhm, H. *et al.* Understanding digitalization's environmental impact: why LCA is essential for informed decision-making. *npj Climate Action* **4**, 41 (2025).
17. Hellweg, S., Benetto, E., Huijbregts, M. A. J., Verones, F. & Wood, R. Life-cycle assessment to guide solutions for the triple planetary crisis. *Nat Rev Earth Environ* **4**, 471–486 (2023).
18. Bertoli, S., Goujon, M. & Santoni, O. The CERDI-seadistance database. Preprint at <https://doi.org/10.5281/zenodo.46822> (2016).
19. Song, S. *et al.* Comparison of vegetable production, resource-use efficiency and environmental performance of high-technology and conventional farming systems for urban agriculture in the tropical city of Singapore. *Science of The Total Environment* **807**, 150621 (2022).
20. Singapore Food Agency. A sustainable food system for Singapore and beyond. *Food for Thought, a digital publication by Singapore Food Agency* <https://www.sfa.gov.sg/food-for-thought/article/detail/a-sustainable-food-system-for-singapore-and-beyond> (2022).
21. International Civil Aviation Organization. *ICAO Carbon Emissions Calculator Methodology Version 2*.
22. Sacchi, R. *et al.* PROspective EnvironMental Impact asSEment (premise): A streamlined approach to producing databases for prospective life cycle assessment using integrated assessment models. *Renewable and Sustainable Energy Reviews* **160**, 112311 (2022).